# A TetR-family transcription factor regulates fatty acid metabolism in the archaeal model organism *Sulfolobus acidocaldarius*

Kun Wang[1], David Sybers[2], Hassan Ramadan Maklad[2], Liesbeth Lemmens[2], Charlotte Lewyllie[2,5], Xiaoxiao Zhou[3], Frank Schult[3], Christopher Bräsen[3], Bettina Siebers[3], Karin Valegård[4], Ann-Christin Lindås[1] & Eveline Peeters[2]

Fatty acid metabolism and its regulation are known to play important roles in bacteria and eukaryotes. By contrast, although certain archaea appear to metabolize fatty acids, the regulation of the underlying pathways in these organisms remains unclear. Here, we show that a TetR-family transcriptional regulator (FadR$_{Sa}$) is involved in regulation of fatty acid metabolism in the crenarchaeon *Sulfolobus acidocaldarius*. Functional and structural analyses show that FadR$_{Sa}$ binds to DNA at semi-palindromic recognition sites in two distinct stoichiometric binding modes depending on the operator sequence. Genome-wide transcriptomic and chromatin immunoprecipitation analyses demonstrate that the protein binds to only four genomic sites, acting as a repressor of a 30-kb gene cluster comprising 23 open reading frames encoding lipases and β-oxidation enzymes. Fatty acyl-CoA molecules cause dissociation of FadR$_{Sa}$ binding by inducing conformational changes in the protein. Our results indicate that, despite its similarity in overall structure to bacterial TetR-family FadR regulators, FadR$_{Sa}$ displays a different acyl-CoA binding mode and a distinct regulatory mechanism.

[1] Department of Molecular Biosciences, The Wenner-Gren Institute, Stockholm University, Svante Arrhenius v. 20C, SE-10691 Stockholm, Sweden. [2] Research Group of Microbiology, Department of Bioengineering Sciences, Vrije Universiteit Brussel, Pleinlaan 2, B-1050 Brussels, Belgium. [3] Molekulare Enzymtechnologie und Biochemie, Biofilm Centre, ZWU, Fakultät für Chemie, Universität Duisburg-Essen, Universitätsstr. 2, 45117 Essen, Germany. [4] Molecular Biophysics, Department of Cell and Molecular Biology, Uppsala University, Uppsala, Sweden. [5] Present address: Laboratory of Cell Genetics, Department of Biology, Vrije Universiteit Brussel, Pleinlaan 2, B-1050 Brussels, Belgium. Correspondence and requests for materials should be addressed to A.-C.L. (email: Ann.Christin.Lindas@su.se) or to E.P. (email: Eveline.Peeters@vub.be)

The phylogenetic classification of archaea as a domain of life distinct from bacteria[1] is supported by the nature of their membrane lipids having isoprenoid-based hydrocarbon chains instead of fatty acids[2]. Despite the absence in archaeal membrane lipids, small amounts of fatty acids and derivatives have been detected in archaeal cells[3–9]. The role of fatty acids for archaeal cellular physiology is not yet clear and a controversial issue of debate[10], although an involvement in the acylation or stabilization of membrane-bound energy-conversion proteins such as rhodopsin or cytochromes has been postulated[10–12]. Many archaeal genomes have extensive sets of typical bacterial-like genes encoding fatty acid synthase type II (FAS-II) complex and β-oxidation enzymes[10,13,14]. An outstanding question is whether these fatty acid metabolism genes perform anabolic or catabolic reactions, or both[9,10]. Given the absence of genes encoding acyl-carrier protein (ACP) or ACP synthase[13], it has been postulated that a β-oxidation pathway might operate in the reverse direction in conjunction with acetyl-CoA C-acetyltransferase enzymes[10]. These are abundantly encoded in archaeal genomes, sometimes in the direct neighborhood of β-oxidation genes[10,14].

Despite the abundance of fatty acid metabolism genes in many genomes, nothing is known about how their expression is regulated in archaea. In contrast, this is well characterized in bacteria, in which a tight regulation of the synthesis and degradation of fatty acids involves multiple transcription regulators that act in response to intracellular fatty acid-related metabolic signals[15]. In Gram-negative bacteria a GntR-family regulator FadR has a dual role by coordinately repressing β-oxidation genes while activating FAS-II genes in response to acyl-CoA molecules[16,17], whereas a TetR-family malonyl-CoA-dependent regulator FabR controls the ratio between mono-unsaturated and saturated fatty acids[18–20]. Gram-positive bacteria such as Bacillus subtilis use an identically named transcription factor FadR that belongs to the TetR family for the acyl-CoA dependent regulation of β-oxidation degradation[21] and a DeoR family member FapR that regulates biosynthesis of saturated fatty acids and phospholipids[22]. The mechanism of action of the bacterial acyl-CoA responsive TetR-like regulator has been unraveled by analysis of apo, ligand-bound, and DNA-bound crystal structures[23–26].

In this work, we focus on characterizing the transcriptional regulation of genes encoding fatty acid metabolism functions in the thermoacidophilic crenarchaeon Sulfolobus acidocaldarius, which is genetically tractable and considered to be a major archaeal model organism[27,28]. S. acidocaldarius has an extensive gene cluster, comprising genes Saci_1103 until Saci_1126, encoding homologs of the three β-oxidation enzymes acyl-CoA dehydrogenase, enoyl-CoA hydratase, and hydroxyacyl-CoA dehydrogenase. Also, genes encoding members of the thiolase superfamily presumably catalyzing the last step of the β-oxidation cycle, i.e., ketoacyl-CoA thiolases as well as acetyl-CoA acetyltransferases were identified within the cluster[10]. In addition, genes encoding lipid degradation functions are present in this genomic region. Concerning these latter functions, Saci_1105 and Saci_1116 code for enzymes that were experimentally shown to display esterase activity[29]. The Saci_1103-Saci_1126 gene cluster also comprises a gene, Saci_1107, encoding a predicted TetR-like transcription factor for which we hypothesized that it might be involved in regulating the expression of the gene cluster. We aim at performing structural, biochemical, genetic and genomic analyses of this regulator, named FadR$_{Sa}$, thereby unveiling the function and mode of action of an acyl-CoA-responsive transcriptional regulator in an archaeal microorganism.

## Results

**FadR$_{Sa}$ structure.** S. acidocaldarius harbors a 30-kb gene cluster consisting of genes encoding enzymes involved in lipid and fatty acid metabolism and a putative regulator (Saci_1107, Fig. 1a). As a first step towards functional characterization of this regulator, we performed a crystallographic analysis of the protein encoded by Saci_1107 (Figure 1b and Table 1). Size exclusion chromatography (SEC) indicated that the purified recombinant protein behaves as a homogenous population of 44-kDa sized dimers (Supplementary Figure 1). The asymmetric unit of the 2.4-Å resolution crystal structure also contains a homodimer with an exclusive alpha-helical structure. Each subunit displays two functional domains: an N-terminal helix-turn-helix (HTH) DNA-binding domain (α1–α3) and a C-terminal domain (α4–α9) of which α8 and α9 stabilize dimerization. The overall Ω-shape structure of the dimer validates its classification as a TetR family member[30].

Although BLAST analyses initially did not reveal which bacterial regulators could be considered as potential functional homologs for the protein encoded by Saci_1107, a superposition revealed structural similarity with the previously characterized TetR-family FadR transcription regulators in Bacillus sp., FadR$_{Bs}$ (RMSD = 4.23 Å) and FadR$_{Bh}$[21,26] (RMSD = 5.88 Å), and Thermus thermophilus, FadR$_{Tt}$[24] (RMSD = 11.85 Å) (Fig. 1c). Conservation is significantly higher for the N-terminal than for the C-terminal domains (Supplementary Table 1) as also confirmed by a structure-based sequence alignment (Fig. 1d). This structural similarity led us to propose to name this protein FadR$_{Sa}$ accordingly.

Upon solving the FadR$_{Sa}$ crystal structure, one of the subunits (subunit B) was found to have additional unassigned electron density in the C-terminal domain. This could be explained by fitting it with an acyl-CoA molecule (Fig. 1b), which was likely derived from Escherichia coli during heterologous overexpression. The best fit was obtained with heptanoyl-CoA. Given the low intracellular abundance of odd-chained acyl-CoA molecules it is possible that a mixture of even-chained short-chain acyl-CoA molecules was present in the ligand binding pockets of different protein molecules packed in the crystal. The unintended cocrystallization of acyl-CoA with FadR$_{Sa}$ (Fig. 1b) suggests that it is a specific ligand of the protein. This further supports the hypothesis that the regulatory role of this transcription factor is connected to acyl-CoA metabolism.

**Genome-wide DNA-interaction map of FadR$_{Sa}$.** As a next step toward unraveling FadR$_{Sa}$ function, we employed chromatin immunoprecipitation (ChIP) in combination with next-generation sequencing (ChIP-seq). A total of 14 significant and reproducible in vivo-associated genomic loci were identified (Fig. 2a and Supplementary Table 2). The two highest enrichments were observed within the Saci_1103-Saci_1126 gene cluster. More specifically, both high-enrichment binding regions were located within the intergenic region of the divergently organized operon encoding the fadR$_{Sa}$ gene itself and a putative esterase-encoding gene (peaks 1 and 2). Within the Saci_1103-Saci_1126 gene cluster, two additional low-enrichment binding regions were observed within the coding sequence of gene Saci_1115 and in the intergenic region separating a divergently encoded β-oxidation operon and a putative transcription factor gene, respectively (peaks 3 and 4). Targeted chromatin immunoprecipitation quantitative polymerase chain reaction (ChIP-qPCR) validated the observed enrichments (Fig. 2b), which were not observed anymore upon deleting fadR$_{Sa}$ (Supplementary Figure 2). All sequences enriched in the ChIP-seq analysis were subjected to a computational binding motif prediction, yielding a 16-base pair

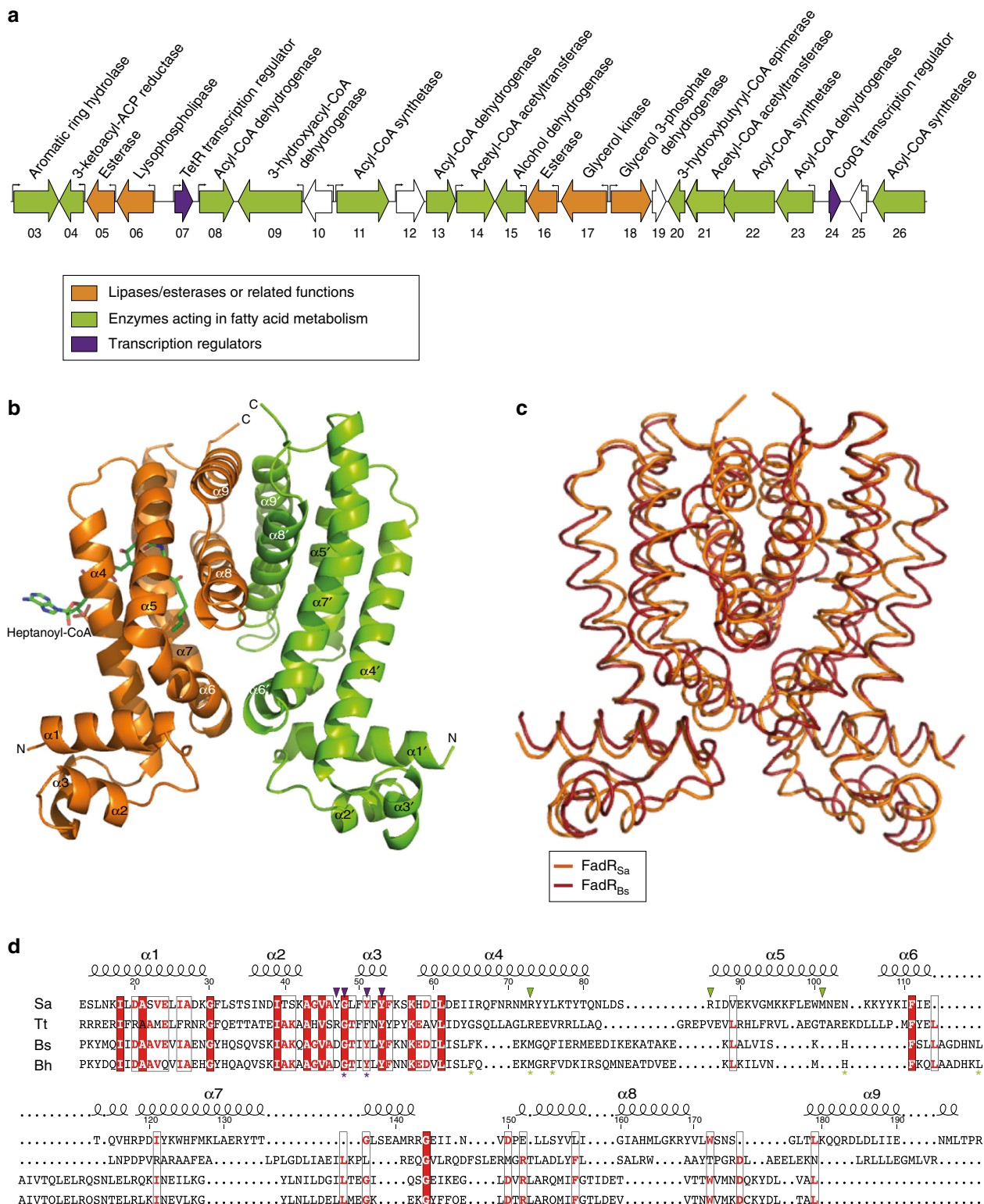

(bp) motif with dyad symmetry that is conserved in 13 of 14 binding regions (Fig. 2c and Supplementary Table 2). Besides the four experimentally identified binding sites, an in silico screening revealed three additional putative binding sites in the gene cluster, of which one is located within the open reading frame (ORF) of *Saci_1106* (Supplementary Figure 3, Supplementary Table 3). Possibly they were not captured in the ChIP-seq analysis but are functional in other conditions.

Electrophoretic mobility shift assays (EMSAs) with DNA probes encompassing the centers of the binding regions verified that the observed ChIP-seq enrichment regions represent direct and specific FadR$_{Sa}$–DNA interactions (Fig. 2d, Supplementary Table 2 and Supplementary Figure 4). Densitometric analysis of EMSA autoradiographs performed with the high-enrichment targets revealed a formation of two electrophoretically distinct FadR$_{Sa}$–DNA complexes with high affinity and positive-binding

**Fig. 1** *S. acidocaldarius* harbors a TetR-family regulator structurally similar to bacterial TetR-like FadR proteins. **a** Genomic organization of the *Saci_1103-Saci_1126* gene cluster. Gene numbers are indicated by displaying the last two digits below each gene arrow. Transcriptional start sites are shown with small arrows, and are based on ref. [37]. **b** Structure of the FadR$_{Sa}$ dimer with indication of the different helices (chain A: α1′–α9′; chain B: α1–α9). The acyl-CoA molecule present in chain B is shown as a stick model. **c** Superposition of the FadR$_{Sa}$ and FadR$_{Bs}$ (PDB 3WHB) structures. **d** Structure-based sequence alignment of TetR-family FadR proteins. The alignment is based on a three-dimensional comparison between FadR$_{Sa}$ from *S. acidocaldarius* (PDB 5MWR), FadR$_{Bs}$ from *B. subtilis* (PDB 3WHB)[25], FadR$_{Bh}$ from *B. halodurans* (PDB 5GP9)[26], and FadR from *T. thermophilus* (PDB 3ANG)[24]. Regions harboring structural and sequence similarity are boxed, with identical amino-acid residues indicated as bold white letters on a red background and functionally equivalent residues indicated in red letters. Secondary structure elements and numbering for FadR$_{Sa}$ are indicated above the sequences. DNA-binding residues targeted for mutagenesis are indicated with purple triangles, ligand-binding residues with green triangles. FadR$_{Bh}$ residues important for DNA binding and ligand binding[26] are indicated below the sequences with purple and green asterisks, respectively

**Table 1 Data collection and refinement statistics (values in parentheses are for outer resolution shell)**

|  | SeMet FadR$_{Sa}$ | FadR$_{Sa}$:DNA | FadR$_{Sa}$: lauroyl–CoA |
|---|---|---|---|
| *Data collection* | | | |
| Space group | $P2_1$ | $P2_12_12_1$ | $P2_1$ |
| *Cell dimensions* | | | |
| a, b, c (Å) | 42.0, 98.8, 56.0 | 54.8, 178.3, 266.7 | 46.0, 91.8, 53.5 |
| α, β, γ (°) | 90.0, 106.4, 90.0 | 90.0, 90.0, 90.0 | 90.0, 112.1, 90.0 |
| Wavelength (Å) | 0.97895 | 0.97625 | 0.97625 |
| Resolution (Å) | 47.2–2.4 | 49.4–3.29 | 45.9–1.90 |
| $R_{merge}$ | 0.046 (0.363) | 0.095 (0.746) | 0.065 (0.727) |
| $I/\sigma(I)$ | 10.34 (1.67) | 12.59 (1.96) | 10.03 (1.07) |
| Completeness (%) | 95.0 (72.9) | 99.5 (95.7) | 94.4 (66.0) |
| Redundancy | 2.2 | 6.8 | 3.3 |
| *Refinement* | | | |
| Resolution (Å) | 47.2–2.4 | 49.4–3.29 | 45.9–1.90 |
| No. of reflections | 72,292 | 276,981 | 100,877 |
| *No. of atoms* | | | |
| Protein | 3075 | 9345 | 4837 |
| Ligand/ion | 56 | 1783 | 122 |
| Water | 9 | 0 | 67 |
| Wilson B-factors (Å²) | 53.5 | 97.4 | 34.0 |
| *R.M.S. deviation* | | | |
| Bond lengths (Å) | 0.009 | 0.008 | 0.002 |
| Bond angles (°) | 0.99 | 0.91 | 1.031 |

cooperativity. In contrast, with the low-enrichment targets only a single complex was formed with lower affinity and without apparent cooperativity (Fig. 2d and Supplementary Figure 4). These findings suggest that for different genomic targets, FadR$_{Sa}$ is capable of using distinct DNA-interaction modes differing in binding stoichiometry.

**Mechanisms of DNA binding**. To further unravel mechanisms of DNA binding by FadR$_{Sa}$, we determined the cocrystal structure of the protein–DNA complex to a resolution of 3.29 Å using a duplex DNA containing the predicted FadR$_{Sa}$ binding motif in the control region of the *fadR$_{Sa}$* gene itself, corresponding to ChIP-seq peak 2 (Table 1 and Fig. 3a, b). The asymmetric unit contained six FadR$_{Sa}$ subunits, organized as three dimers, and two DNA duplex molecules thus representing a nonbiological assembly (Fig. 3a), although the protein–DNA molecular interactions within this structure are representative of the biologically relevant complexes (see below, DNA-binding stoichiometry of FadR$_{Sa}$).

In each FadR$_{Sa}$ subunit in the cocrystal structure, residues of the recognition helix α3 and the α2–α3 loop interact with the

major groove of DNA with the establishment of an extensive number of contacts (Fig. 3b, Supplementary Note 1 and Supplementary Data 1). Base-specific contacts mainly consist of hydrophobic interactions between FadR$_{Sa}$ residues Tyr47, Leu49, Tyr51, and Phe52 and methyl groups of thymines (Fig. 3c) similar as in other TetR-like regulators[26,30] (Supplementary Figure 5), in addition to electrostatic interactions between Gly48 and the N7 group of guanines. The role of these residues for DNA binding was further investigated by performing site-directed alanine substitution and analyzing the mutant proteins in EMSA (Supplementary Figure 6a). FadR$_{Sa}^{Y47A}$, FadR$_{Sa}^{Y51A}$, and FadR$_{Sa}^{Y53A}$ are all negatively affected in DNA-binding affinity and cooperativity. With FadR$_{Sa}^{G48A}$, no DNA binding was observed at all demonstrating that Gly48 is a crucial residue (Supplementary Figure 6a). Besides protein–DNA contacts, a weak electrostatic protein–protein contact was also observed between Asn37 residues of FadR$_{Sa}$ dimers bound on different sides of the DNA helix (Fig. 3d and Supplementary Note 1).

**DNA-binding stoichiometry of FadR$_{Sa}$**. To dissect the stoichiometric nature of the electrophoretically distinct FadR$_{Sa}$–DNA complexes, SEC was performed with the different molecular species (Fig. 4a). With a homogenous population of dimers in solution (Fig. 4a and Supplementary Figure 1) and the FadR$_{Sa}$–*Saci_1123* complex B having an apparent molecular weight (MW) measured in SEC of 140 kDa that is only minimally exceeding that of free DNA (119 kDa), it can be concluded that FadR$_{Sa}$ binds the *Saci_1123* operator as a single dimer. The observation that the relative mobility in EMSA of FadR$_{Sa}$–*Saci_1123* complex B is highly similar to that of the FadR$_{Sa}$–*fadR$_{Sa}$* complex B1 (Fig. 4b), led us to postulate that the transitional FadR$_{Sa}$–*fadR$_{Sa}$* complex B1 has a stoichiometry similar as for the sole FadR$_{Sa}$–*Saci_1123* complex. In contrast, the dominantly formed complex B2 with the *fadR$_{Sa}$* operator has a larger apparent MW (179 kDa) (Fig. 4a): it can be assumed that the apparent MW attributed by the FadR$_{Sa}$ protein itself is similar for measurements of free protein and of FadR$_{Sa}$–DNA complexes and that the FadR$_{Sa}$–*fadR$_{Sa}$* complex B2 has a stoichiometry that is twice as large as that of complex B1, thus harboring two dimers. SEC experiments with lower protein:DNA molar ratios indicate that the entire amount of FadR$_{Sa}$ in the preparation is capable of binding DNA (Supplementary Figure 7). This excludes the possibility that a subpopulation of the protein is in a ligand-induced state lacking DNA-binding activity as suggested by the observation of acyl-CoA cocrystallizing with the protein in the apo crystal structure (Fig. 1b), assuming that acyl-CoA binding causes DNA dissociation like in bacterial FadR regulators.

Next, footprinting experiments were performed for the FadR$_{Sa}$–DNA complexes B1 and B2 observed in EMSAs with *fadR$_{Sa}$* and quasi-identical *Saci_1106* operator probes (representing ChIP-seq peaks 1 and 2, respectively) (Fig. 4c and Supplementary Figure 8). Chemical "in gel" Cu–phenanthroline footprinting demonstrated that for the *fadR$_{Sa}$* operator probe,

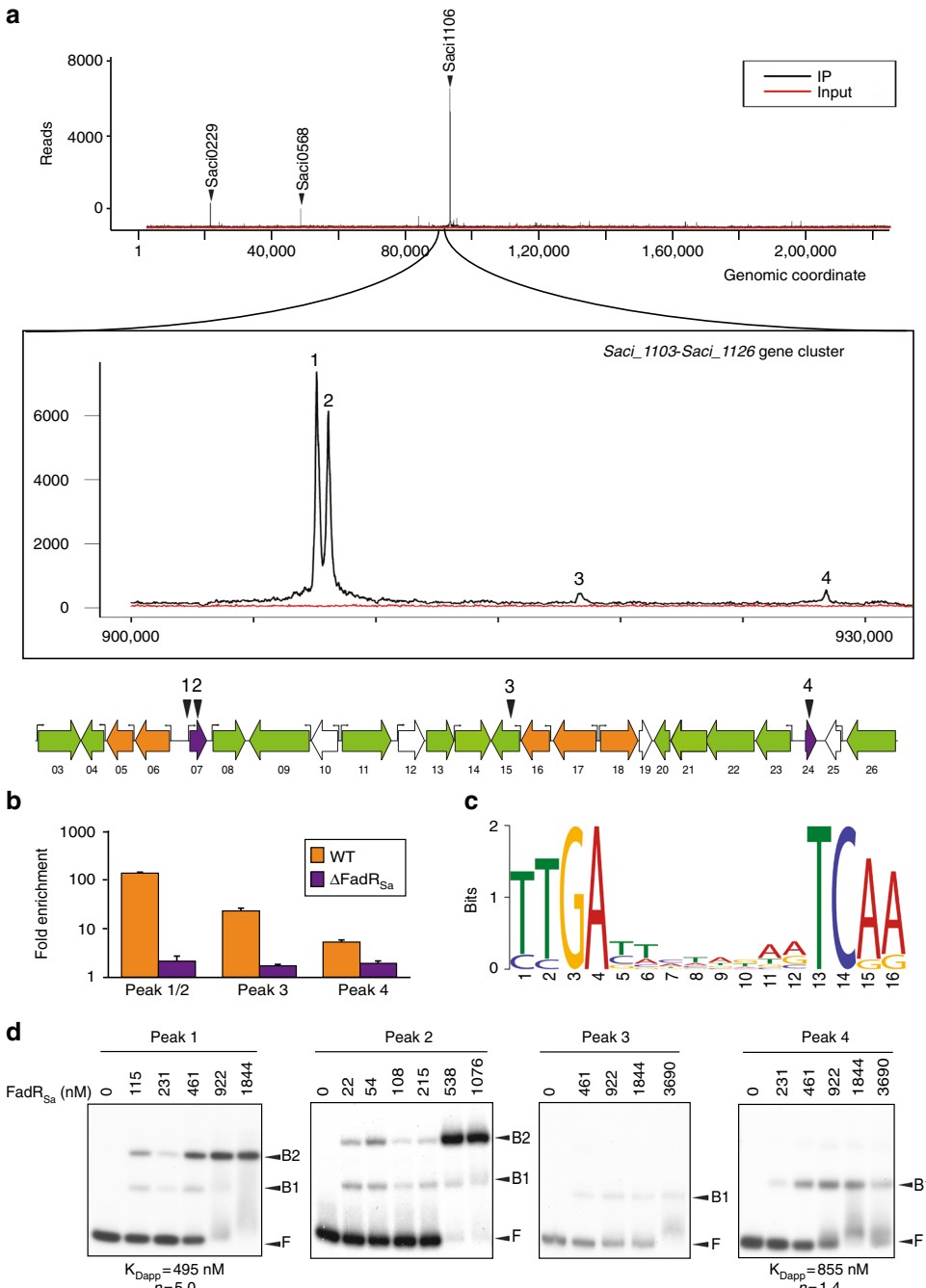

**Fig. 2** FadR$_{Sa}$ interacts with four genomic loci in the *Saci_1103-Saci_1126* gene cluster. **a** Overview of the genomic binding profile of FadR$_{Sa}$ as monitored by ChIP-seq (IP = immunoprecipitated sample). A zoomed image of this profile is shown for the genomic region encompassing the *Saci_1103-Saci_1126* gene cluster (corresponding to genomic coordinates 903,000–932,000), with indication of the four clearly visible peaks (numbered 1–4). Below the profile, a schematic representation of the genomic organization of the *Saci_1103-Saci_1126* gene cluster is shown with indication of the ChIP-seq peak summit locations and of the transcription start sites as detected in the transcriptomic analysis in ref. [37]. **b** ChIP-qPCR experiment with targeted quantification of enrichment for peaks 1 and 2 (given their close proximity to each other, these are assayed within a single fragment representing the *Saci_1106/Saci_1107* intergenic region), peak 3 and peak 4. Fold enrichment is expressed relative to a genomic region within the ORF of *Saci_1336* that was shown not be bound by FadR$_{Sa}$ in the ChIP-seq profile. Error bars represent standard deviations of biological duplicates. **c** Sequence logo of the FadR$_{Sa}$ binding motif representing MEME predictions of ChIP-seq enriched sequences. **d** Electrophoretic mobility shift assays of FadR$_{Sa}$ binding to radiolabeled DNA probes of about 500 bp representing the ChIP-seq peaks identified in the *Saci_1103-Saci_1126* gene cluster (see panel (**a**)). Molar protein concentrations are shown above each autoradiograph, whereas populations of free DNA (F) and FadR$_{Sa}$-bound DNA (B1 and B2) are indicated with an arrowhead. Apparent $K_D$ and Hill coefficient (*n*) calculations are based on densitometric analysis of free DNA bands followed by binding curve analysis (Supplementary Figure 4)

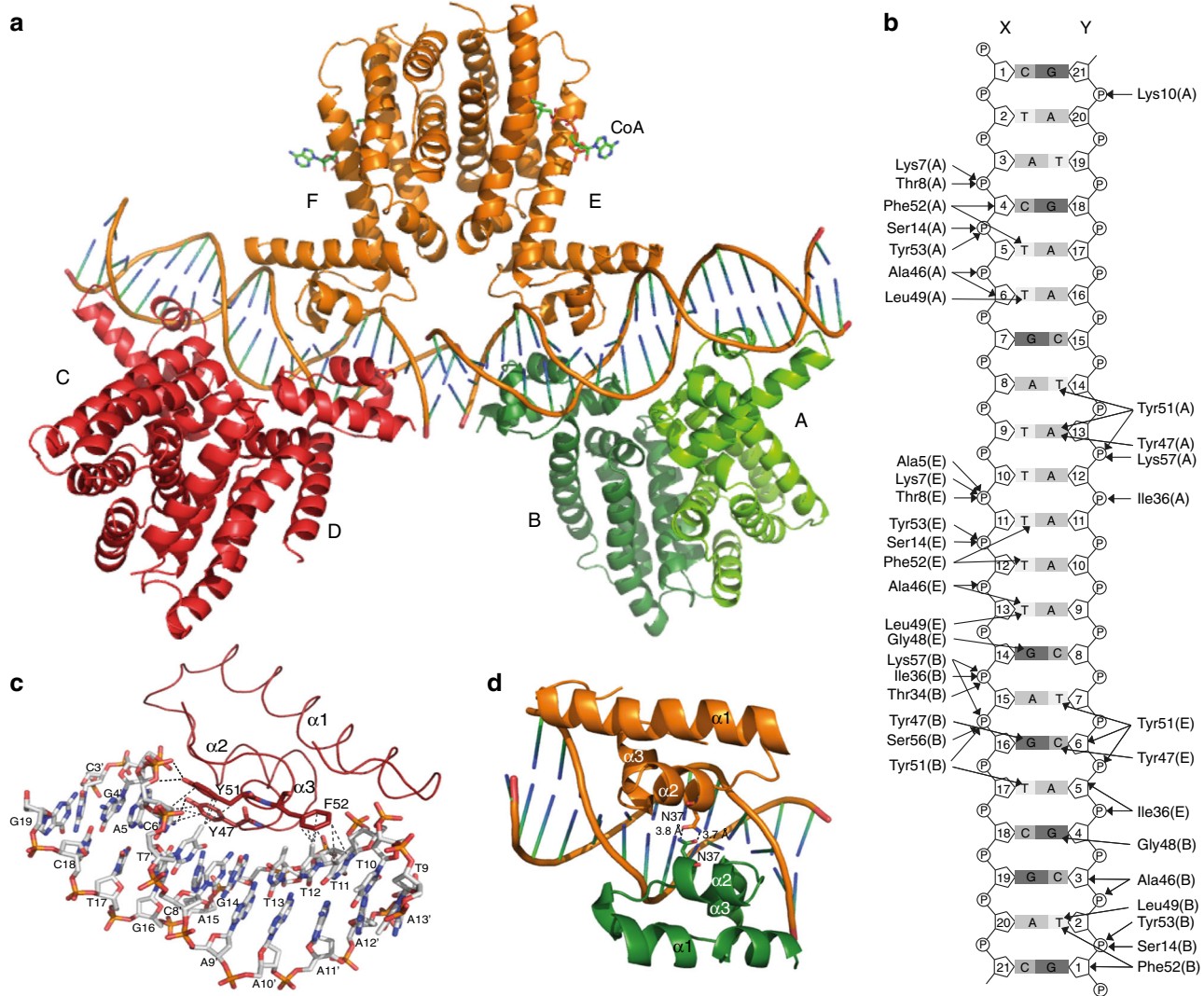

**Fig. 3** A FadR$_{Sa}$–DNA cocrystal structure reveals an important role for hydrophobic interactions. **a** Cartoon representation of the cocrystal structure of FadR$_{Sa}$ in complex with the operator site in the *fadR$_{Sa}$* promoter region. Different FadR$_{Sa}$ subunits are labeled A–E, each dimer is colored differently with subunits A and B belonging to the same dimer while being displayed in a darker and lighter green, respectively, for the sake of clarity. **b** Detailed map of interactions identified in the FadR$_{Sa}$–DNA cocrystal structure, summarizing Supplementary Data 1. **c** Zoom of the interface between subunit E and the X–Y DNA duplex, with emphasis on interactions established by residues Tyr47, Tyr51, and Phe52. Hydrogen bonds are indicated by black and hydrophobic interactions by gray dashed lines. Bases are labeled with those belonging to chain Y being displayed with a prime. **d** Zoom of the interaction between subunits E (in orange) and B (in green). Weak electrostatic interactions are indicated with dashed lines

both complexes are characterized by a similar protection zone roughly restricted to the predicted binding motif with a small stretch of additional protection extending upstream of the motif in complex B2 (Fig. 4c). This upstream extension was also observed in DNase I footprinting experiments for both operator probes (Fig. 4c and Supplementary Figure 8). In contrast, the protection zones observed in footprinting experiments with the *Saci_1123* operator probe (representing ChIP-seq peak 4) are smaller and only correspond to the binding motif (Supplementary Figure 9). These results support the notion that in the FadR$_{Sa}$–*Saci_1123* complex B and the FadR$_{Sa}$–*fadR$_{Sa}$* complex B1 a single-FadR$_{Sa}$ dimer interacts with the predicted semipalindromic binding motif. A second dimer interacts with the upstream (left) side of the inverted repeat in FadR$_{Sa}$–*fadR$_{Sa}$* complex B2, suggesting a dimer-of-dimer interaction mode similar to that observed for a bacterial subclass of TetR-like regulators prototyped by QacR in *Staphylococcus aureus*, in which two overlapping inverted repeats are bound by dimers located on opposite sides of the DNA helix[31].

To relate the molecular interactions and binding architecture in the FadR$_{Sa}$–DNA cocrystal structure with the biologically relevant stoichiometries, we mapped the contacts on the *fadR$_{Sa}$* and *Saci_1123* operator sequences that could be hypothesized to exist based on the cocrystal structure (Fig. 5a). The presence of a G–C bp 11 positions upstream of the symmetrical C–G in the pseudopalindromic site appears to be the sole explanation of the preference of a second FadR$_{Sa}$ dimer to establish an interaction with the upstream (left) and not downstream (right) side of the pseudopalindromic site. Similarly, the absence of purine–pyrimidine and pyrimidine–purine bps on appropriate positions up- and downstream of the inverted repeat in the *Saci_1123* operator appears to explain why only a single dimer is bound. Based on the cocrystal structure, the G base of this G–C bp in the *fadR$_{Sa}$* operator can be assumed to be contacted by Gly48 that is crucial for DNA binding (Fig. 5b and Supplementary Figure 6a) with the dimer-of-dimer-complex architecture reflecting that of the architecture of the AB dimer and subunit E of the DE dimer interacting with a single-DNA duplex in the

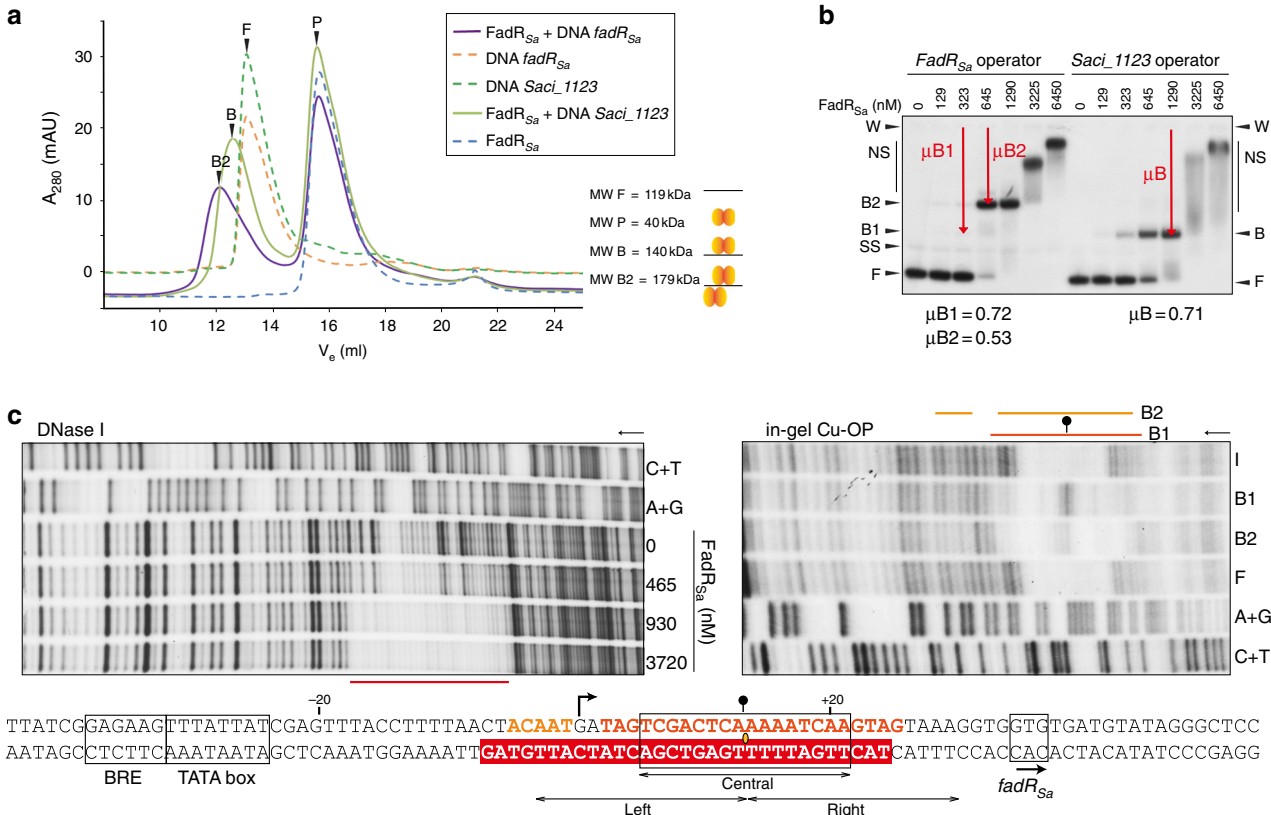

**Fig. 4** FadR$_{Sa}$ interacts in different stoichiometric binding modes with different operators. **a** Size-exclusion chromatography experiment of FadR$_{Sa}$–DNA complexes with 45-bp DNA fragments. Totally, 0.2 nmol DNA and 8 nmol FadR$_{Sa}$ was used resulting in a 40:1 FadR$_{Sa}$:DNA molar ratio upon mixing. Apparent molecular weights (MWs) based on $K_{av}$ calculations are mentioned for the peaks representing free DNA (F), FadR$_{Sa}$ protein (P) and FadR$_{Sa}$–DNA complexes B and B2. **b** Electrophoretic mobility shift assay (EMSA) monitoring interaction with probes representing the *fadR$_{Sa}$* and *Saci_1123* operator. Molar protein concentrations are indicated and populations of single-stranded DNA (SS), free DNA (F), and FadR$_{Sa}$-bound DNA (B, B1, and B2) are indicated with an arrowhead. "W" corresponds to the well position. Relative mobilities $\mu$ are defined as the distance between the well position and complexed DNA divided by the distance between the well position and free DNA. **c** Autoradiographs of DNase I (bottom-strand labeled DNA) and "in-gel" Cu-phenantroline (Cu-OP) (top-strand labeled DNA) footprinting experiments analyzing FadR$_{Sa}$ binding to a probe representing peak 2 in the *Saci_1106-Saci_1107* intergenic region (*fadR$_{Sa}$* operator). A + G and C + T denote purine- and pyrimidine-specific Maxam–Gilbert sequencing ladders, respectively. I denotes input DNA (taken from a protein-free lane in the EMSA), F denotes the population of free DNA, while B1 and B2 denote different FadR$_{Sa}$:DNA complex populations in accordance with the notation in the corresponding EMSA (second autoradiograph in Fig. 2d). The I and F samples generated a sequence-dependent cleavage profile. Protected zones are indicated with a horizontal line while a hyperreactivity site is pointed out with a ball-and-stick symbol. Below the autoradiographs, the nucleotide sequence is shown with a summary of the observed protection zones and hyperreactivity effect, the latter indicating FadR$_{Sa}$-induced DNA bending and being more pronounced for B1 than for B2. White letters in a red background represent the protection zone observed in DNase I footprinting, whereas orange letters represent the protection zone observed in Cu-OP footprinting. The predicted pseudopalindromic FadR$_{Sa}$ recognition site is boxed with indication of the center of dyad symmetry. The transcriptional start site, indicated with an arrow, is based on observations in ref. [37]

cocrystal structure (Fig. 3a). This reasoning is underscored by the observation that the introduction of a G–C and C–G bp at the indicated positions of the *Saci_1123* operator causes the formation of two instead of one nucleoprotein complex (Fig. 5c). Given that adenines also have an N7 group, the base specificity of the Gly48–guanine interaction is possibly explained by an indirect readout of the preceding thymidine or cytosine residue in the light of YpG base pair steps being more prone to unstacking and commonly involved in sequence-specific protein–DNA interactions in a combined direct and indirect readout[32]. In addition, the mutation of Asn37 causes a diminished cooperativity in the formation of the dimer-of-dimer complex B2 with the mutated *Saci_1123* operator (Fig. 5c), proving the involvement of this residue in a protein–protein interaction. This supports the notion that established FadR$_{Sa}$–DNA contacts are similar in the complex in solution as in the portion of the cocrystal structure harboring dimer AB and subunit E of dimer EF and that the relative

positioning of the two dimers in the biologically relevant complex is similar to that of dimers AB and EF in the cocrystal structure.

In conclusion, while bacterial TetR proteins are subdivided in two classes depending on whether they employ a dimer or dimer-of-dimer DNA-binding mode[26,30], the archaeal FadR$_{Sa}$ regulator is capable of using both interaction modes depending on the operator sequence. A Gly48–guanine interaction and Asn37-mediated protein–protein contacts, of which the latter have never before been observed for bacterial dimer-of-dimer binding TetR-like proteins, assist in the dimer-of-dimer interaction mode.

**Determination of the FadR$_{Sa}$ regulon.** To infer whether or not the observed genomic binding of FadR$_{Sa}$ leads to transcriptional regulation, a comparative transcriptomic analysis was performed for the *fadR$_{Sa}$* deletion mutant *versus* the isogenic WT strain using an RNA-seq approach (Fig. 6a, Supplementary Note 2,

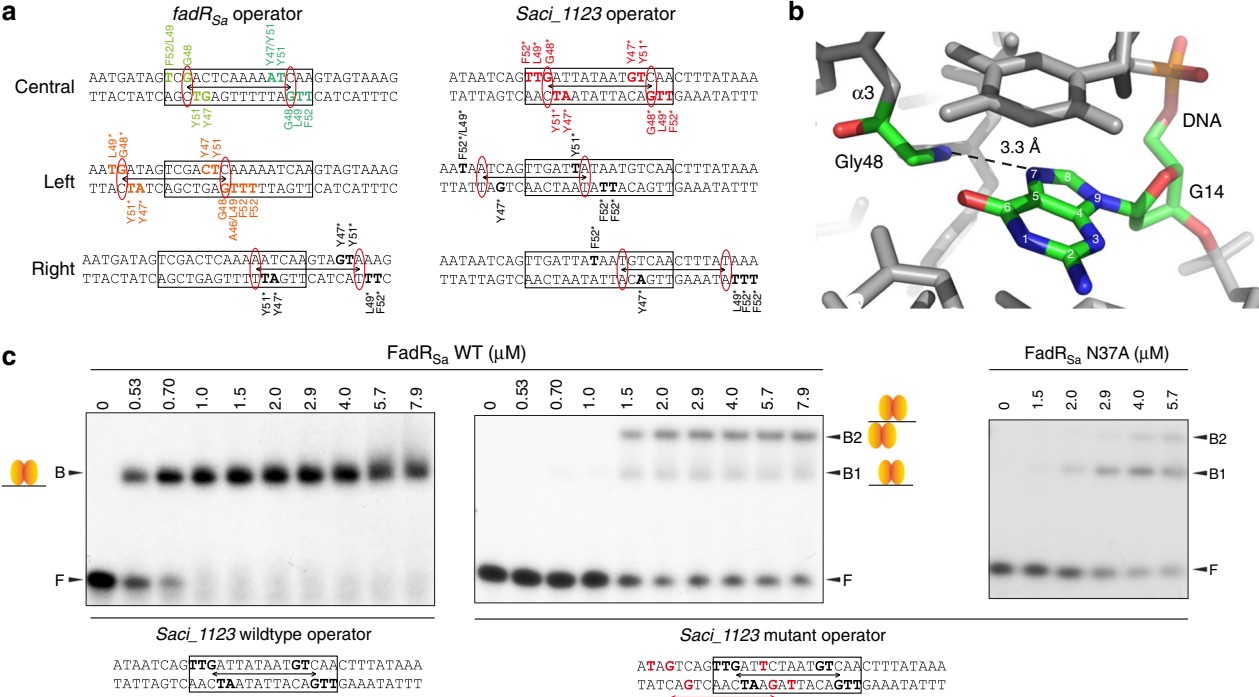

**Fig. 5** Specific protein–DNA contacts determine stoichiometry of the interaction. **a** Map of observed and hypothesized interactions for the *fadR_Sa* and *Saci_1123* operators. Colored base and amino acid residues without an asterisk represent contacts identified in the FadR_Sa–DNA cocrystal structure (using the same color code for the subunits as in panel (**a**)). Colored residues with an asterisk are hypothesized to be established in the natural operators. Black residues in bold are those that theoretically could support binding in case a GC (AT) or CG (TA) would have been present on the crucial Gly48-interacting positions, which are indicated with red ovals. **b** Zoom of the Gly48–G14 interaction in the interface between subunit E and the X–Y DNA duplex in the FadR_Sa–DNA cocrystal structure. **c** EMSAs with 45-bp DNA fragments representing a wildtype or mutated variant of the *Saci_1123* operator region, performed with wildtype or N37A mutant FadR_Sa. Molar protein concentrations are shown above the autoradiograph, whereas populations of free DNA (F) and FadR_Sa–bound DNA (B, B1, and B2) are indicated with an arrowhead, as well as with schematic representations of the binding stoichiometry

Supplementary Figure 10, Supplementary Table 4 and Supplementary Data 2). The deletion of *fadR_Sa* did not affect cell morphology and growth in a medium containing sucrose and NZamine as carbon and energy sources (Supplementary Figure 11). RNA-seq analysis revealed that thirteen genes are differentially expressed, which all belong to the *Saci_1103-Saci_1126* gene cluster. Moreover, as confirmed by quantitative reverse transcriptase PCR (qRT-PCR), all other genes of the gene cluster appear to be expressed at slightly higher levels in the mutant strain as well (Fig. 6a–b and Supplementary Data 2). We therefore conclude that FadR_Sa is a local repressor of the entire *Saci_1103-Saci_1126* gene cluster, which is predicted to harbor genes for a complete β-oxidation pathway (Fig. 6a).

All but one of these FadR_Sa binding sites are located at too large distances (>130 bp) from their corresponding promoters to hypothesize a classical repression mechanism that involves direct interaction with the components of the basal transcription machinery. As an exception, the *fadR_Sa* control region harbors a binding site just downstream of the transcription start site. For this target, it is shown that FadR_Sa binding stimulates the interaction with basal transcription factors TATA binding protein (TBP) and transcription factor B (TFB) (Supplementary Figure 12), pointing to a direct repression mechanism occurring at later stages of transcription initiation than during TBP and TFB recruitment.

Besides the local regulon, several other genes, including an operon encoding a putative sulfate reduction pathway and cytochrome-encoding genes, were found to have slightly lower expression levels in the *ΔfadR_Sa* strain pointing to an indirect activation effect (Fig. 6b, Supplementary Note 2 and Supplementary Data 2). The suggestion of this effect being indirect is

corroborated by the prediction of only a limited number of putative FadR_Sa binding sites in the genomic regions surrounding the downregulated genes, which are characterized by relative high *p* values ($>1.00E{-}05$) (Supplementary Table 3) and which were not captured by ChIP-seq. Furthermore, these indirect regulatory effects hint at a reversely correlated link between the fatty acid metabolism catalyzed by the enzymes encoded in the *Saci_1103-Saci_1126* gene cluster on one hand and sulfur metabolism and cytochrome-containing membrane complexes on the other hand.

The observed transcriptional regulation of the *Saci_1103-Saci_1126* gene cluster strongly suggests that FadR_Sa is implicated in the regulation of fatty acid and lipid metabolism. Since it was previously observed that simultaneous deletion of both esterase-encoding genes in the gene cluster (*Saci_1105* and *Saci_1116*) led to a phenotype lacking the ability to perform tributyrin hydrolysis[29], we performed a similar phenotypic assay with the *fadR_Sa* deletion mutant (Supplementary Figure 13a). Despite the higher expression levels of both esterase genes in the *fadR_Sa* deletion mutant (Fig. 6a–b), we did not observe a difference in time-dependent halo formation upon growth on tributyrin (Supplementary Figure 13a). In contrast, upon growing *S. acidocaldarius* in a liquid medium containing hexanoate as a sole carbon and energy source, the *fadR_Sa* deletion mutant displayed a significantly higher growth rate in exponential growth phase with respect to the isogenic WT strain (doubling times $T_d$s of 20.5 and 26.3 h, respectively; Fig. 6c). As this effect was not observed during growth on the shorter-chain butyrate (Supplementary Figure 13b), it correlates to fatty acid chain length. These experiments support that *S. acidocaldarius* is capable of degrading fatty acids to sustain growth and that this catabolic metabolism may be at least partly catalyzed by enzymes encoded in the

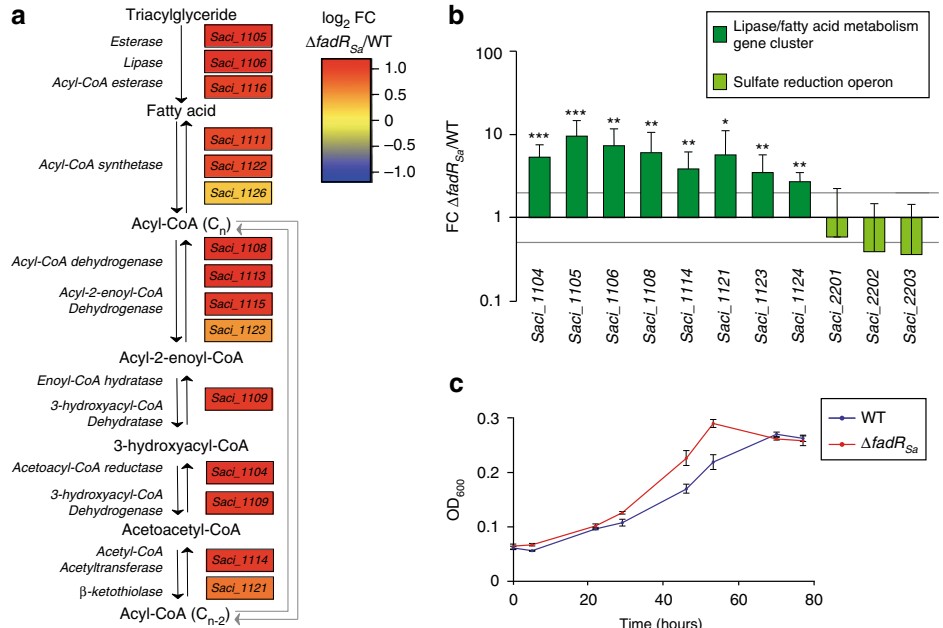

**Fig. 6** FadR$_{Sa}$ exerts a repression on the transcription of the catabolic fatty acid metabolism/lipase gene cluster. **a** Reconstruction of the lipid degradation and fatty acid β-oxidation pathway with indication of the putative functions of genes in the *Saci_1103-Saci_1126* gene cluster predicted to be involved in this metabolic pathway. This is based on genomic annotations. For each of these genes, differential expression in *S. acidocaldarius* MW001 Δ*fadR$_{Sa}$*, as measured by RNA-seq analysis, is shown in a heat map format expressed as the log$_2$ of the fold change expression in the Δ*fadR$_{Sa}$* versus its isogenic wildtype strain. **b** Relative gene expression as determined by qRT-PCR for a subset of the genes of the *Saci_1103-Saci_1126* gene cluster and for genes belonging to the sulfate reduction operon. Gray lines represent a fold change of 2 and 0.5, respectively. Error bars represent biological variation for triplicates (standard deviations). An asterisk indicates a *p*-value between 0.05 and 0.01, a double asterisk between 0.01 and 0.001, and a triple asterisk smaller than 0.001, as determined in a statistical *t* test. **c** Growth curves of MW001 and MW001 Δ*fadR$_{Sa}$* strains in Brock medium containing 2 mM hexanoate as sole carbon source. Values are averages of four biological replicates with error bars representing standard deviations. Doubling times ($T_{d}$s) were calculated by modeling the exponential section of the growth curves. Representative curves are shown for multiple independently performed experiments

*Saci_1103-Saci_1126* gene cluster. Furthermore, the FadR$_{Sa}$ regulator represses this catabolic fatty acid metabolism as its deletion, thereby causing a derepression of the gene cluster, results in a faster growth rate (Fig. 6c). The observation of this difference can be explained by hexanoate only causing a partial FadR$_{Sa}$-mediated derepression given the relative short-chain length of these acyl-CoA molecules (see below, "FadR$_{Sa}$–ligand interactions").

**FadR$_{Sa}$–ligand interactions**. Next, we prepared FadR$_{Sa}$ crystals in the presence of lauroyl–CoA (C12:0-CoA) and solved the FadR$_{Sa}$–lauroyl–CoA cocrystal structure to a resolution of 1.90 Å (Table 1). In contrast to the initial structure showing an acyl-CoA derivative bound to only one of the subunits, in this structure, both subunits of the dimer harbor a lauroyl–CoA molecule (Supplementary Figure 14a). The orientation of the ligand-binding pockets within the protein and the binding mode of the ligand is completely different in FadR$_{Sa}$ as compared to the characterized bacterial FadR proteins[24–26] (Fig. 7a). In contrast to the ligand entering the pocket from within the dimer interface as in bacterial FadR regulators, in the FadR$_{Sa}$ structure the ligand enters the protein from the outer surface of the protein completely opposite to the dimer interface. Further, for each ligand-binding pocket only a single FadR$_{Sa}$ subunit is involved in ligand interaction in contrast to two subunits in the bacterial FadR regulators. Consequently, ligand conformation is different and the acyl chain has a rather straight conformation in FadR$_{Sa}$ while it is bent in FadR$_{Bs}$ (Supplementary Figure 14b).

Upon zooming into the ligand-binding pocket, a large number of specific lauroyl–CoA–FadR$_{Sa}$ interactions are observed (Fig. 7b

and Supplementary Data 3). Whereas the adenine moiety is located on the outside of α4 and appears not to be contacted by the protein, the remainder of the lauroyl–CoA molecule enters the protein in between α4 and α7 with the establishment of electrostatic interactions between polar residues (a.o. Arg73, Lys77, Arg86, and Arg132) and the CoA moiety, especially with the CoA phosphate groups (Fig. 7b). Upon comparison of residue conformations in the DNA-bound and lauroyl–CoA structures, it became apparent that the orientation of the α5 residue Met101 is significantly altered (Supplementary Figure 15a). The lauroyl chain is deeply buried into a tunnel-like binding pocket formed in between helices α4–α7 and is entirely surrounded by hydrophobic residues such as Phe68, Phe97, Phe111, and Phe126. Although the nature of these ligand-interaction residues (polar residues for CoA-interactions, hydrophobic residues for side chain interactions) is similar as in bacterial TetR-like FadR regulators, they are not homologous as shown on a structure-based sequence alignment (Fig. 1d). Furthermore, FadR$_{Sa}$ does not contain a hydrophilic patch in the acyl-binding pocket similarly to the *Bacillus* counterparts in which it affects ligand-binding specificity[25,26].

EMSAs demonstrated that acyl-CoA molecules, but not acetyl-CoA, CoA and free fatty acids, strongly abrogate FadR$_{Sa}$–DNA complex formation with the extent of the abrogation effect correlating with the length of the acyl chain (Fig. 7d and Supplementary Figure 16a). Competition assays confirmed that the addition of acetyl-CoA or hexanoyl-CoA does not affect sensitivity of the protein to oleoyl-CoA and thus that the inhibition effect reflects binding specificity (Supplementary Figure 16b). Alanine substitution of the CoA-interacting residues

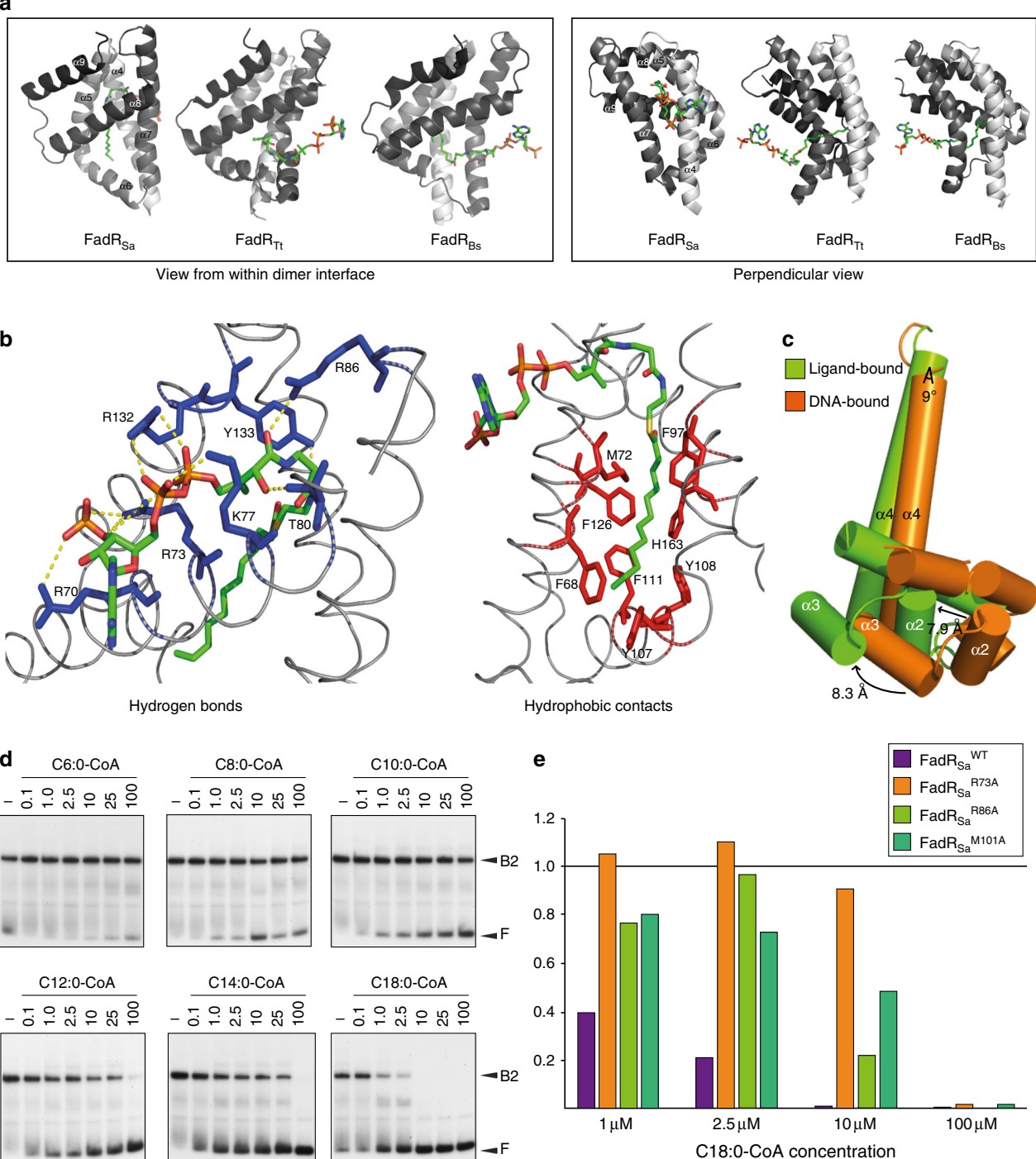

**Fig. 7** Acyl-CoA disrupts FadR$_{Sa}$–DNA complexes proportional to acyl chain length. **a** Structural comparison of ligand-binding modes of lauroyl–CoA-bound FadR$_{Sa}$ (PDB: 6EL2), lauroyl–CoA-bound FadR$_{Tt}$ (PDB: 3ANG)[24], and lauroyl–CoA-bound FadR$_{Bs}$ (PDB: 3WHB)[25]. Only the C-terminal domains are shown in cartoon representation, with α-helices α4–α9 colored from light to dark gray; the lauroyl–CoA molecule is shown as a stick model. Two different viewpoints are shown: a view from within the monomer-monomer interface in the dimer and a view perpendicular to that. **b** Close-up view of the ligand-binding pocket with indication of FadR$_{Sa}$ residues involved in establishing polar contacts with lauroyl–CoA (in blue) and of residues involved in hydrophobic interactions (in red). **c** Schematic representation of a close-up view of helices α1–α4 with indication of relative conformational differences. A detailed structural superimposition is shown in Supplementary Figure 15b. **d** Electrophoretic mobility shift assays (EMSAs) demonstrating the effect of acyl-CoA on the FadR$_{Sa}$–DNA interaction using a 154-bp probe representing the *Saci_1106* binding site. Acyl-CoA concentrations are shown above each autoradiograph in μM. Populations of free DNA and complexed DNA are indicated with F and B2, respectively. The intermediate complex B1 is hardly formed due to the cooperativity of the interaction. **e** Graphical representation of ligand response measured in EMSAs performed with ligand-binding mutants FadR$_{Sa}^{R73A}$, FadR$_{Sa}^{R86A}$, and FadR$_{Sa}^{M101A}$ (Supplementary Figure 6b). The *Y*-axis represents the fraction of bound DNA with respect to a protein-free control lane

Arg73 and Arg86 confirmed their importance for the ligand response (Fig. 7e and Supplementary Figure 6b). Likewise, alanine mutation of the allosterically altered Met101 residue desensitizes the protein to oleoyl-CoA. In conclusion, acyl-CoA binds the regulator thereby causing dissociation of FadR$_{Sa}$–DNA complexes with the affinity and extent of the effect correlating to the acyl chain length.

**Molecular mechanism of ligand response.** To learn more about the allosteric regulatory mechanism employed by FadR$_{Sa}$, we compared the ligand-bound and DNA-bound structures (Fig. 7c and Supplementary Figure 15b). Both structures were super-imposed with an RMSD of 1.01 Å (Supplementary Figure 15b). Subtle differences were noted in the relative orientation of the HTH motifs within a dimer with the binding of lauroyl–CoA causing the distance between the two α3 helices to be enlarged from an average 37.0–43.2 Å (measured as the Cα–Cα distance of Tyr51 residues located in α3). As a consequence, the increased distance between the α3 helices makes the dimeric FadR$_{Sa}$ conformation suboptimal for interaction in consecutive major groove segments. Besides the α3 helix, the α2 helix is shifted by an average distance of 7.9 Å and the α4 helix is displaced with an angle of 9° (Fig. 7c). Intriguingly, one of the three dimers in the FadR$_{Sa}$–DNA cocrystal structure appeared to have a ligand-bound conformation distinct from the other two dimers; this is reflected by the distance between the two α3 helices being 45.3 Å for this central dimer EF versus an average of 37.0 Å for the flanking dimers (Fig. 3a; Supplementary Note 3). In conclusion, ligand binding allosterically opens up the dimer thereby causing it to dissociate from the DNA, similarly as the mechanism observed for FadR$_{Bh}$[26].

**Occurence of FadR in archaea.** FadR$_{Sa}$ is not restricted to *S. acidocaldarius* but is also represented in all other *Sulfolobus* species, in other Crenarchaeota belonging to Sulfolobales (*Acidianus* spp.) and Thermoproteales (*Cuniculiplasma divulgatum*), and also in species belonging to Euryarchaeota (*Thermoplasmatales*) and in the recently discovered Marsarchaeota[33] (Fig. 8). These organisms have in common that they are all thermophiles, some with a (facultative) aerobic metabolism, others with an anaerobic metabolism. Conservation of residues that are involved in DNA or ligand binding indicate that these proteins are FadR$_{Sa}$ orthologs with similar functional characteristics (Fig. 8a). Moreover, as is the case for FadR$_{Sa}$, several of these homologs are encoded in genomic environments abundant in genes coding for enzymes involved in fatty acid metabolism or for enzymes with lipase functions (Fig. 8b). Although gene synteny is not strictly conserved, the extent of some of these gene clusters, especially in other *Sulfolobus* species, suggests the potential existence of similar FadR-mediated acyl-CoA responsive repression. This hypothesis is supported by the prediction of putative FadR binding sites in the neighborhood of *fadR* promoters and at distant locations, either in intergenic regions or in ORFs, for the gene clusters in other *Sulfolobus* species (Fig. 8b).

## Discussion
It is well-established that archaea harbor typical bacterial-like transcription regulators[34,35], which are proposed to result from shared ancestry as well as from extensive horizontal gene transfers, especially from bacteria to archaea[36]. FadR$_{Sa}$, of which we show that it displays structural similarities with bacterial TetR-like FadR regulators, is an archaeal member of the widespread prokaryotic TetR family. Despite these similarities, there are pronounced differences between the archaeal FadR regulator and the bacterial counterparts which point to a complete absence of shared ancestry. For example, the acyl-CoA binding function of FadR$_{Sa}$ appears to have arisen through convergent evolution with respect to the bacterial regulators. In contrast to FadR$_{Bs}$ and FadR$_{Tt}$, in which the crystal structure revealed medium-chain lauroyl–CoA in one of the two dimeric subunits[24,25], we observed the presence of a short-chain acyl-CoA in the native FadR$_{Sa}$ structure. This difference likely reflects different acyl-CoA-binding specificities and might be explained by the absence of a hydrophilic patch in the part of the ligand-binding pocket that surrounds the first 10–12 carbon atoms of the acyl chain, as observed for the *Bacillus* counterparts[25,26]. As a consequence, FadR$_{Sa}$ has a different ligand specificity, which is expected to have consequences for the biological function and to reflect different biological roles of fatty acids for cellular physiology in bacteria and archaea. Interestingly, the observation of an inverse correlation between the expression of the *Saci_1103-Saci_1126* gene cluster and that of cytochrome-encoding genes further supports the suggested function of fatty acids stabilizing membrane complexes in archaea[10].

FadR$_{Sa}$ uses two distinct DNA-binding modes operator-dependently with a Gly–guanine interaction being the crucial determinant and that dimer-of-dimer complex formation with the quasi-identical high-affinity *fadR$_{Sa}$* and *Saci_1106* operators occurs in a cooperative manner with the transitional formation of a dimer-bound complex in which FadR$_{Sa}$-mediated DNA bending is more pronounced than in the dimer-of-dimer complex. By binding to a total of only four binding sites in the *Saci_1103-Saci_1126* gene cluster, FadR$_{Sa}$ is capable of repressing transcription of the entire 23-gene cluster containing at least 17 transcription units[37]. Furthermore, with the exception of the autoregulatory binding site that is located just downstream of the TTS and that is expected to result in repression through direct interaction with the basal transcription initiation machinery, the other binding sites are located at least 130 bp upstream of the corresponding promoters, which is a noncanonical position as compared to most previously characterized archaeal repressors[38]. FadR$_{Sa}$ thus employs a nonparadigmatic repression mechanism that could be hypothesized to be dependent on long-range interactions. The observation of cocrystallization of FadR$_{Sa}$ with two individual DNA duplexes (Fig. 3a), as was also observed for the bacterial TetR member CgmR in *Corynebacterium glutamicum*[39], further supports the possibility that the regulator is capable of co-associating with different DNA segments.

The finding that FadR$_{Sa}$ represses the *Saci_1103-Saci_1126* gene cluster and that it is responsive to acyl-CoA molecules acting as inducers in vitro strongly suggests that intracellularly present acyl-CoA molecules cause a derepression and thus higher transcriptional expression of the gene cluster in vivo. The observation that deletion of the regulator causes cells to display a faster growth on hexanoate as sole energy and carbon source supports that the β-oxidation enzymes encoded in this gene cluster minimally have a degradation function; this is in line with the logic behind the regulatory strategy. A catabolic function of the β-oxidation enzymes is also in agreement with the function of the co-regulated esterase enzymes encoded by *Saci_1105* and *Saci_1116*, which enable cells to grow on lipids[29]. Fatty acid oxidation adds to the chemoorganotrophic capabilities of *Sulfolobus* spp. that appear more important than the originally described chemolithotrophic sulfur-oxidizing metabolism[40]. A full picture of the functioning of fatty acid metabolism in *Sulfolobus*, and whether the enzymes encoded in the *Saci_1103–1126* function only the catabolic or also anabolic direction, awaits the biochemical and genetic characterization of the enzymes. An intriguing hypothesis has been put forward stating that fatty acid metabolism enzymes in archaea do not have a catalytic bias but are instead regulated by the relative substrate and product

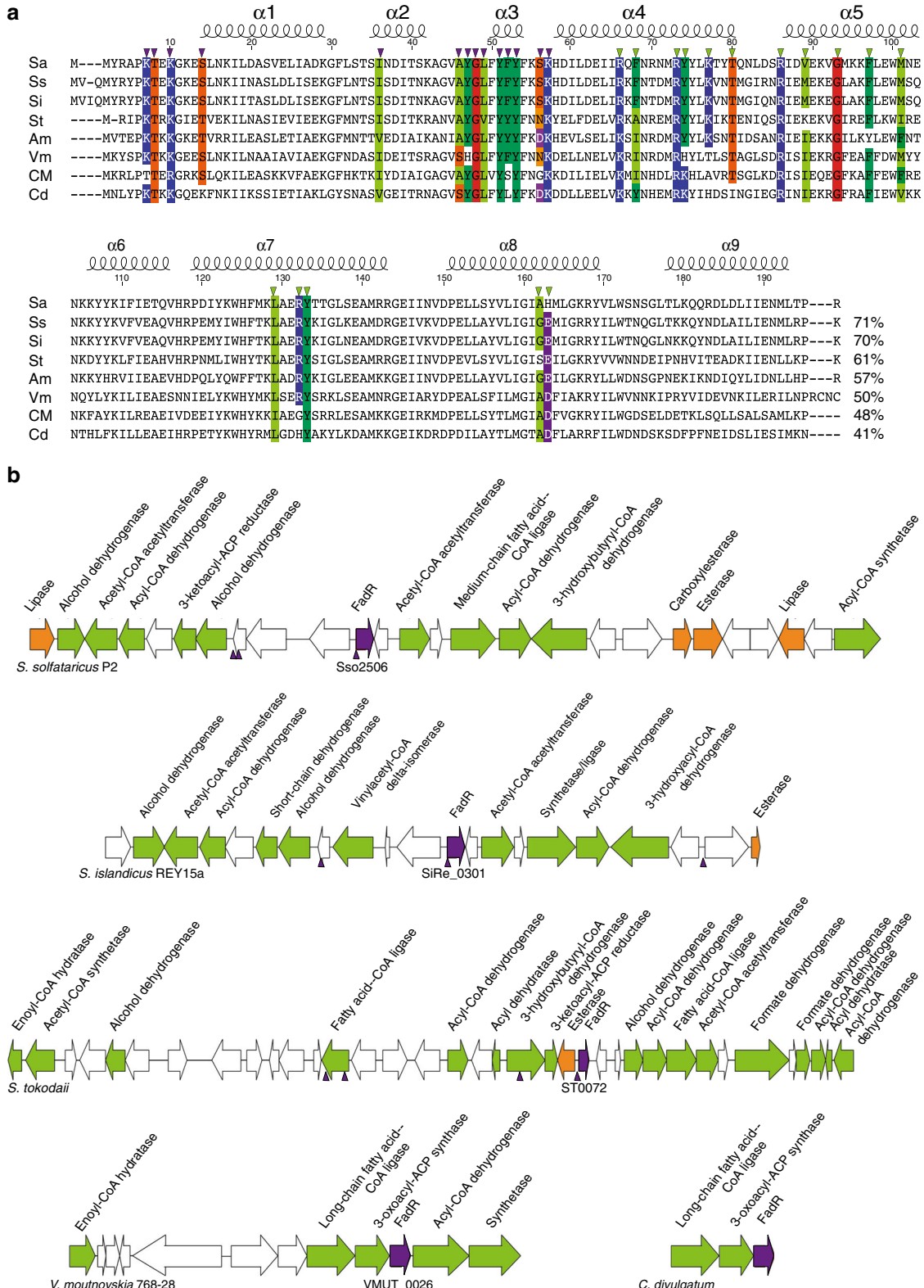

concentrations[9,10]. Possibly, acyl-CoA-responsive FadR$_{Sa}$-mediated induction of the expression of the *Saci_1103-Saci_1126* gene cluster is only a single element in a more complex regulatory system in which other regulatory mechanisms are also in place to enable the fine-tuned expression and activity of promiscuous enzymes in response to relative concentrations of a variety of fatty acid-related metabolic signals. This postulated regulatory complexity might reflect the employment of the same enzymes for catabolic and anabolic reactions instead of the use of distinct dedicated pathways as in bacteria and eukaryotes. Furthermore, based on the occurrence of FadR$_{Sa}$ orthologs in other thermophilic archaea, such as *Thermoplasmatales* and Marsarchaeota, it

**Fig. 8** FadR_Sa homologs are found in other archaeal genomes. **a** Multiple sequence alignment, prepared with t-coffee[68], of different archaeal FadR_Sa homologs identified through BLAST analysis. Sa = FadR_Sa *Sulfolobus acidocaldarius*, Ss = AAK42639.1 *Sulfolobus solfataricus* P2, Si = PVU77113.1 *Sulfolobus islandicus*, St = WP_010978011.1 *Sulfolobus tokodaii*, Am = ARM75525.1 *Acidianus manzaensis*, Vm = WP_013603407.1 *Vulcanisaeta moutnovskia*, CM = PSN82693.1 Candidatus Marsarchaeota G1 archaeon OSP_D, Cd = WP_077076373.1 *Cuniculiplasma divulgatum*. Indication of secondary structure elements and of position numbers is with respect to the FadR_Sa sequence. Colors indicate conservation of amino acid residues that are involved in DNA (purple triangles) or ligand (green triangles) interactions, based on FadR_Sa cocrystal structures. Sequence identities with FadR_Sa are mentioned behind the sequence alignment. **b** Schematic depiction of gene organization in the genomic neighborhood of selected FadR_Sa homologs. Color coding is as follows: green = genes encoding enzymes involved in fatty acid metabolism, orange = genes encoding lipases/esterases, purple = genes encoding transcriptional regulators. Gene numbers are mentioned for the FadR_Sa homologs. The *Sulfolobus spp.* gene cluster sequences were subjected to a binding motif prediction using RSAT[69] with a position weight matrix based on FadR_Sa binding site sequences. Predicted binding sites are indicated with a blue triangle

can be assumed that this type of transcriptional regulation is not restricted to *Sulfolobus* spp.

## Methods

**Microbial strains and growth conditions.** *S. acidocaldarius* strains MW001 and its derivatives were cultivated at 75 °C in Brock basal salts medium[40] supplemented with 0.2 (w/v)% sucrose, 0.2 (w/v)% NZamine and 10 µg ml$^{-1}$ uracil. For ChIP experiments, a 200 ml *S. acidocaldarius* DSM639 culture was grown at 80 °C in Brock basal salts medium supplemented with 0.1% tryptone. The pH of the medium was adjusted to 3.5 by addition of sulfuric acid. For growth experiments in the presence of fatty acids, precultures of *S. acidocaldarius* MW001 and its derivatives were first grown in Brock medium supplemented with 0.2 (w/v)% NZamine followed by multiple transfers to Brock medium supplemented with 2 mM butyrate or hexanoate. Cultivations of the third and fourth generation were considered to lack residual NZamine and to be representative for growth on fatty acids as sole carbon source. For growth on plates, Brock medium was solidified by the addition of 0.6 (w/v)% gelrite, 10 mM MgCl$_2$ and 3 mM CaCl$_2$. To detect the hydrolysis of tributyrin, cells were inoculated on tributyrin-containing plates composed of 1.4% gelrite dissolved in Brock basal salt medium supplemented with 20 mM MgCl$_2$ × 6H$_2$O, 6 mM CaCl$_2$, 0.2% (w/v) NZamine, 0.4% (w/v) dextrine and 1% (v/v) tributyrin[29]. Cavities were made into the plates using a 5 ml pipette tip followed by addition of 20 µl of cell culture grown up to an optical density measured at 600 nm (OD$_{600}$) of 0.7. The plates were incubated up to 7 days.

*E. coli* strains DH5α and Rosetta DE3 were used for the propagation of plasmid DNA constructs and heterologous protein overexpression, respectively, and were grown at 37 °C in lysogeny broth (LB) medium supplemented with 50 µg ml$^{-1}$ ampicillin (DH5α) or with 50 µg ml$^{-1}$ ampicillin and 34 µg ml$^{-1}$ chloramphenicol (Rosetta (DE3)). An overview of all strains used in this work is given in Supplementary Table 5.

A markerless in-frame *fadR_Sa* deletion strain (*S. acidocaldarius* Δ*fadR_Sa*) was generated of the uracil auxotrophic *S. acidocaldarius* strain (MW001) using the classical "pop-in pop-out" method with a suicide disruption vector containing up- and downstream flanking regions and the gene-of-interest region besides the *pyrEF* selection marker genes (pSVA406xΔ*fadR*)[28]. An overview of all plasmid vectors and oligonucleotide sequences is given in Supplementary Table 6 and Supplementary Data 4, respectively. Successful deletion of the gene was confirmed by PCR analysis and sequencing of genomic DNA (gDNA) (Supplementary Figure 2).

**Protein expression and purification.** The *fadR_Sa* coding region was amplified by PCR from *S. acidocaldarius* gDNA and cloned into a pET45b expression vector using BamHI and HindIII restriction sites (pET45bx*fadR_Sa*) resulting in an N-terminally His-tagged construct. Site-directed mutagenesis was performed with the overlap PCR mutagenesis approach[41] using pET45bx*fadR_Sa* as a template and complementary mutagenic primers to remove an NdeI restriction site in the *fadR_Sa* ORF with a silent mutation. This enabled cloning in the NdeI/XhoI sites of pET24a yielding a C-terminally His-tagged construct (pET24ax*fadR_Sa*Ndenull). For the construction of N37A, Y47A, G48A, Y51A, Y53A, R73A, R86A, and M101A variants of FadR_Sa, site-directed mutagenesis was performed in an identical approach using pET24ax*fadR_Sa*Ndenull as a template.

Heterologous overexpression of the recombinant FadR_Sa proteins was accomplished in *E. coli* Rosetta (DE3) by growing a culture until reaching an OD$_{600}$ between 0.6 and 0.7, incubating the cells on ice during 30 min and inducing gene expression by the addition of 0.4 mM isopropyl β-D-1-thiogalactopyranoside (IPTG). Subsequently, the culture was further incubated at 37 °C during 20 h, pelleted by centrifugation, resuspended in binding buffer (20 mM sodium phosphate pH 7.4, 0.5 M NaCl, 20 mM imidazole) and lysed by sonication. Finally, the cell extract was subjected to heat treatment (80 °C during 30 min) followed by centrifugation at 23,000×g during 45 min. Purification of His-tagged FadRSa was performed with immobilized metal ion affinity chromatography by applying the supernatant to a His GraviTrap system (GE Healthcare) equilibrated with binding buffer (20 mM sodium phosphate (pH 7.4), 0.5 M NaCl and 20 mM imidazole). The column was washed with binding buffer where after the protein was eluted with 3 × 1 ml of 20 mM sodium phosphate (pH 7.4), 0.5 M NaCl and 500 mM imidazole. N- and C-terminally His-tagged proteins behave similar in DNA-binding assays.

For the preparation of selenomethionine (SeMet)-substituted FadR_Sa protein, cells were allowed to grow overnight at 37 °C in 2 ml 2 × YT medium containing 50 µg ml$^{-1}$ ampicillin. The overnight culture was used to inoculate 50 ml M9 minimal medium (50 mM Na$_2$HPO$_4$, 3 g l$^{-1}$ KH$_2$PO$_4$, 0.5 g l$^{-1}$ NaCl, 1 g l$^{-1}$ NH$_4$Cl) and growth was continued at 37 °C until an OD$_{600}$ of 0.5 was reached. Thereafter, 50 ml of the culture was added to 700 ml M9 medium supplemented with 2 mM MgSO$_4$, 0.1 mM CaCl$_2$, 0.4% glucose, and 50 µg ml$^{-1}$ ampicillin, and continue growth at 37 °C until an OD$_{600}$ of 0.4–0.6. Following l-amino acids were then added: Lys, Phe, and Tyr (100 mg l$^{-1}$), Leu, Ile, and Thr (50 mg l$^{-1}$) and l-selenomethionine (Acros Organics; final concentration of 60 mg l$^{-1}$). At an OD$_{600}$ of 0.6–0.8, protein expression was induced by adding 0.5 mM IPTG (isopropyl β-D-1-thiogalactopyranoside) followed by overnight incubation. SeMet-substituted protein was purified according to the same procedure as described above using a pET45bx*fadR_Sa* harboring strain. All FadR_Sa protein preparations were essentially pure, as judged by SDS-polyacrylamide gel electrophoresis (PAGE) and by SEC (Supplementary Figure 1).

The ORFs of TBP (Saci_1336) and TFB1 (Saci_0866) were PCR-amplified, digested and ligated into pET30a (Novagen) using the restriction exonucleases NdeI and XhoI and transformed into *E. coli* Rosetta 2 (DE3). Cells harboring pET30ax*tbp* were grown until reaching an OD$_{600}$ of approximately 0.6, followed by an induction with 1 mM IPTG and further growth for 4 h at 37 °C. Cells were harvested by centrifugation and resuspended in lysis buffer (25 mM Tris-HCl pH 7.5, 300 mM NaCl) supplemented with protease inhibitor cocktail (Roche) and disrupted via French pressure cell (Thermo Electron Corporation, USA) for three passages at 12,000 psi followed by ultracentrifugation (30,000×g during 45 min). A heat treatment (80 °C during 15 min) was again followed by ultracentrifugation and TBP protein was further purified by anion exchange chromatography using a ResourceQ column (GE Healthcare) with a salt gradient up to 1 M NaCl and SEC on a HiLoad superdex 26/60 75 prep grade column (GE Healthcare) using 25 mM Tris-HCl pH 7.5, 300 mM NaCl. Similarly, *E. coli* Rosetta 2 (DE3) cells expressing C-terminally hexahistidine-tagged TFB1 were grown in medium supplemented with 10 mM MgCl$_2$ and 100 µM ZnSO$_4$, induced at an OD$_{600}$ of approximately 0.6 by adding 0.2 mM IPTG and followed by further growth at 23 °C overnight. Cells were harvested by centrifugation and subsequently resuspended in modified N-buffer (25 mM Tris-HCl pH 7.5, 10 mM MgCl$_2$, 100 µM ZnSO$_4$, 1 mM tris(2-carboxyethyl)phosphine (TCEP)) supplemented with 1 M NaCl and protease inhibitor. After cell disruption via French pressure cell (three passages at 12,000 psi), ultracentrifugation (30,000×g during 45 min) and heat treatment at 75 °C for 15 min the cleared lysate was applied to a Ni-TED column (Macherey and Nagel) for affinity purification. Modified N-buffer containing 300 mM NaCl was used for equilibration and washing, the bound target protein was eluted with the same buffer containing 250 mM imidazole. Pure TFB1 protein was finally obtained by performing SEC with a HiLoad superdex 26/60 200 prep grade gel filtration column (GE Healthcare). TBP and TFB1 were used in EMSAs (Supplementary Figure 12).

**Crystallization and data collection.** Crystallization of SeMet-derivated FadR_Sa was performed at 20 °C using the hanging-drop vapor diffusion method by mixing equal volumes of protein solution (3 mg ml$^{-1}$) and reservoir solution consisting of 20% (w/v) PEG3350, 0.2 M sodium nitrate and 0.1 M Bis–Tris propane, pH 8.5. Appropriately sized crystals were obtained after 6–8 weeks. The crystals belong to space group P2$_1$, with unit-cell parameters $a = 41.9$, $b = 98.7$, $c = 55.9$ Å, and $\beta = 106.4°$, and two molecules per asymmetric unit, giving a Matthews coefficient of 2.51 Å$^3$ Da$^{-1}$ and 51% solvent content. The complexes of FadR_Sa:DNA and FadR_Sa:lauroyl–CoA were obtained by cocrystallization of FadR_Sa with a 21-bp duplex DNA and 1 mM lauroyl–CoA (lithium salt), respectively. Prior to data collection, crystals were soaked in a cryo-solution containing 20% glycerol, 10% (w/v) PEG3350, 0.1 M sodium nitrate and 0.05 M Bis–Tris propane, pH 8.5 followed by immediate flash-cooling in liquid nitrogen.

High-resolution X-ray data (Table 1) were collected at 100 K at European Synchroton Radiation Facility (ESRF) beamlines ID23-1 (SeMet-substituted FadR_Sa) and ID29 (FadR_Sa:DNA and FadR_Sa:lauroyl–CoA).

**Structure determination and refinement.** The structure of FadR_Sa was determined using the SAD method with selenomethionine-substituted protein. Diffraction data were processed and scaled using the *XDS* program package[42]. A set of

5% of the reflections was set aside and used to calculate the quality factor $R_{free}$[43]. The structure was solved using AutoSol in PHENIX[44]. Refinement was performed with PHENIX altered with manual rebuilding in O[45]. The structure was refined to $R_{fac} = 20.3\%$ and $R_{free} = 25.3\%$, respectively (Table 1). The structure of the dsDNA and lauroyl–CoA complexes were solved by molecular replacement with Phaser[46], using the SeMet-subsituted FadR$_{Sa}$ structure as a model. All structures were evaluated using wwPDB Validation Server[47]. Refinement statistics are presented in Table 1. The coordinates and structure factors have been deposited in the PDB database with accession codes 5MWR, 6EN8, and 6EL2. PDBsum[48] was used to identify protein–DNA and protein–ligand interactions, supplemented with a manual inspection employing PyMOL[49]. All figures displaying protein structures were prepared with PyMOL[49].

**Chromatin immunoprecipitation**. ChIP was performed by growing *S. acidocaldarius* DSM639 to early exponential growth phase (an $OD_{600}$ between 0.2 and 0.3) and adding formaldehyde to the culture to a final concentration of 1%[50,51]. After a 5-min incubation, glycine was added to a final concentration of 125 mM. Fixed cells were harvested by centrifugation at 8000×*g* during 10 min and the pellet was resuspended in 3 ml IP buffer (50 mM Hepes-KOH pH 7.5, 150 mM NaCl, 1 mM EDTA, 1% Triton X-100, 0.1% sodium deoxycholate, 0.1% SDS, protease inhibitor cocktail (Roche Applied Science)). Subsequently, cells were sonicated on ice until DNA fragments were obtained with an average size around 250 bp. After centrifugation during 20 min at 17,000×*g*, 100 μl of the sheared DNA-containing supernatant was kept apart to use as input control and the remaining sample was divided into two aliquots. One aliquot was incubated with anti-FadR$_{Sa}$ antibody coated M-280 Sheep Anti-Rabbit Dynabeads (Invitrogen) and the other was incubated with pre-immune serum coated Dynabeads, which served as a non-specific binding control (mock control). The bead-antibody complexes were prepared by mixing 80 μl Dynabeads with either FadR$_{Sa}$-specific antibodies (produced by immunizing a rabbit with purified recombinant FadR$_{Sa}$ (Innovagen)) or rabbit pre-immune serum. Precipitation reactions were performed according to manufacturer's instruction. After overnight incubation at 4 °C, the Dynabeads were collected and the captured DNA was eluted and purified by using the iPURE DNA extraction kit (Diagenode) according to the manufacturer's instruction.

DNA purified from ChIP, input and mock samples were sequenced ($1 \times 51$ bp) by a Miseq sequencer (Illumina) at ScilifeLab, Stockholm, Sweden. Sequence reads were mapped to the *S. acidocaldarius* DSM639 genome (NC_007181.1) with Burrows–Wheeler Aligner (BWA 0.7.10)[52] using default settings and MACS2 (2.1.0)[53] was employed for peak calling. To generate sufficient sequencing reads, reads of two mock samples were combined before performing the analysis. The ChIP-seq experiment was done in biological duplicate and only peaks called in both experiments were retained; this was followed by a manual curation. Finally, ChIP-seq results were visualized by IGV version 2.3.59[54]. DNA sequences of enriched regions were extracted by BEDTools' getfasta function[55] and subjected to a binding motif search with MEMEsuite (4.10.0)[56]. The FIMO tool of MEMEsuite was used for the prediction of binding motifs in other genomic regions.

For ChIP-qPCR, 20-ml cultures of *S. acidocaldarius* MW001 and *S. acidocaldarius* MW001Δ*fadR$_{Sa}$* were crosslinked and harvested in mid exponential growth phase ($OD_{600}$ of about 0.4) and ChIP was performed as described above. Primers were designed with Primer3 Plus software[57] (Supplementary Data 5) and were chosen to amplify fragments around the ChIP-seq peak summit regions and with a length between 150 and 200 bp. qPCR was performed with a My-iQ™ Single color Real-time PCR system (Bio-Rad) and GoTaq qPCR Master Mix (Promega) was done with thermal cycling conditions: 10 min at 94 °C and 40 cycles of 30 s at 94 °C and 30 s at 60 °C. Fold-enrichment calculations were performed with the $2^{-\Delta\Delta Ct}$ method[58] using an irrelevant genomic region (the ORF of Saci_1336) as a nonbinding reference. Cultures were assayed in biological duplicate and the *S. acidocaldarius* Δ*fadR$_{Sa}$* strain was used as mock experiment.

**Electrophoretic mobility shift and footprinting assays**. $^{32}$P-labeled DNA was prepared by 5′-end-labeling of oligonucleotides using [γ-$^{32}$P]-ATP (Perkin Elmer) and T$_4$ polynucleotide kinase (Thermo Scientific). Each of these labeled oligonucleotides were then used together with a non-labeled oligonucleotide (Supplementary Data 4) in a PCR reaction with *S. acidocaldarius* gDNA as a template or, in case of 45-bp probes, in a hybridization reaction with the nonlabeled reverse complementary oligonucleotide. Labeled DNA fragments were subsequently purified by polyacrylamide gel electrophoresis. EMSAs were performed[59] with approximately 0.1 nM $^{32}$P-labeled DNA probe and an excess of nonspecific competitor DNA. Dimethylsulfoxide was used as a solvent to dissolve acyl-CoA, but did not affect the protein–DNA interaction (Supplementary Figure 16). Binding reactions were prepared in binding buffer (20 mM Tris-HCl (pH 8.0), 1 mM MgCl$_2$, 0.1 mM dithiothreitol (DTT), 12.5% glycerol, 50 mM NaCl, 0.4 mM EDTA) and allowed to equilibrate at 37 °C prior to electrophoresis on 6% acrylamide gels in TEB buffer (89 mM Tris, 2.5 mM EDTA, and 89 mM boric acid).

DNase I footprinting was performed by the method of Galas and Schmitz[60] in the same binding buffer as used for EMSA and with Maxam–Gilbert treated samples as sequencing ladders[61]. "In gel" Cu–phenantroline (OP) chemical footprinting experiments, enabling to analyze distinct populations of nucleoprotein complexes separately, were performed by performing EMSA as described above and immersing an entire EMSA acrylamide gel in 200 ml of 10

mM Tris (pH 8.0)[59]. To initiate chemical cleavage reactions, 20 ml of solution A (2 mM OP, 0.45 mM CuSO$_4$) was added followed by 20 ml of solution B (58 mM mercaptopropionic acid). After 15 min, reactions were quenched by the addition of 20 ml 30 mM neocuprine hydrate. This mixture was allowed to equilibrate during 5 min, after which the gel was thoroughly rinsed with distilled H$_2$O and exposed to an autoradiograph film. Different DNA populations were excised from the gel, recovered by precipitation and analyzed by denaturing acrylamide gel electrophoresis with Maxam–Gilbert treated samples as sequencing ladders[61].

**Size-exclusion chromatography**. For stoichiometry experiments of FadR$_{Sa}$–DNA complexes, SEC was performed on a Superdex 200 Increase10/30 GL column with an ÄKTA FPLC system (GE Healthcare Life Sciences). A total of 4–40 nM FadR$_{Sa}$ protein was mixed with 0.2–1 nM 45-bp DNA fragments encompassing the *fadR$_{Sa}$* operator or *Saci_1123* operator, respectively, which were prepared before by hybridization. After an incubation of the reaction mixtures in 20 mM Na$_2$HPO$_4$ (pH 7.4), 150 mM NaCl during 25 min at 37 °C, they were loaded onto the column with the same buffer as mobile phase buffer. Calibration for MW calculation was performed with ribonuclease A (13.7 kDa), carbonic anhydrase (29 kDa), con-albumin (75 kDa), and ferritin (440 kDa), all from a Gel Filtration Calibration Kit (GE Healthcare Life Sciences).

**RNA-sequencing analysis**. Total RNA was prepared from duplicate MW001 and MW001Δ*fadR$_{Sa}$* cultures in early exponential growth phase at an $OD_{600}$ between 0.2 and 0.3 using a miRNeasy Mini Kit (Qiagen). Libraries were prepared with a TruSeq Stranded mRNA Library Prep Kit (Illumina). Paired-end ($2 \times 125$ bp) RNA sequencing was performed using a Hiseq2500 system (Illumina) at SciLifeLab, Stockholm, Sweden. Sequence reads were first trimmed to remove sequencing adapters by cutadapt 1.9.1[62] and reads shorter than 20 nt were discarded. Processed reads were then mapped to the *S. acidocaldarius* DSM639 genome (NC_007181.1)[63] with Tophat 2.0.12[64]. For each gene, the FPKM was calculated with Cufflinks 2.2.1[65]. Finally, read counts were obtained by the featureCounts function in the Subread package 1.5.0[66] and only genes having at least one count in all samples were used for differential gene expression analysis with DESeq2[67].

**Quantitative RT-PCR**. RNA was extracted at an $OD_{600}$ between 0.2 and 0.3 by stabilization with RNA Protect Reagens (Qiagen) and by using an SV Total RNA Isolation System (Promega). Residual gDNA was removed by treatment with TURBO DNase (Ambion Life Technologies). cDNA was prepared from 1 μg RNA with an iScript™ Select cDNA Synthesis Kit (Bio-Rad). All qRT-PCR oligonucleotides (Supplementary Data 4) were designed with Primer3 Plus software[57]. qRT-PCR analysis was performed in a Bio-Rad iCycler using SYBR Green Master Mix (Bio-Rad) for amplification and detection. Each reaction mixture contained approximately 10 ng of template and 200 nM of each primer in a total volume of 25 μl. The temperature program was as follow: 10 min at 94 °C and 40 cycles of 30 s at 94 °C, 30 s at 60 °C[50]. Relative expression ratios were calculated using the delta–delta $C_t$ method[58] for biological triplicates and by normalizing with respect to the *tbp* reference gene. Data were statistically analyzed by performing a *t* test with the software package Prism 6.0 (GraphPad).

**Reporting Summary**. Further information on experimental design is available in the Nature Research Reporting Summary linked to this article.

## Data availability
All crystal structures presented in this work have been deposited in the Protein Data Bank (PDB) and are available with accession codes 5MWR, [has been superseded with 6EL2] (native FadR$_{Sa}$ structure), 6EL2 (lauroyl–CoA-bound FadR$_{Sa}$ structure) and 6EN8 (DNA-bound FadR$_{Sa}$ structure). All raw data for the ChIP-seq and RNA-seq studies presented in this work have been deposited in the Gene Expression Omnibus (GEO) databank with accession codes GSE108039 and GSE108018, respectively.

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

## Acknowledgements

We are grateful to Daniël Charlier for assistance with footprinting experiments and critical reading of the manuscript. We also would like to acknowledge support from Science for Life Laboratory, the National Genomics Infrastructure (NGI) and Uppmax in Stockholm/Uppsala, Sweden, for providing assistance in massive parallel sequencing and computational infrastructure. Research in the laboratory of E.P. was supported by the Research Council of the Vrije Universiteit Brussel (start-up funds) and by the Research Foundation Flanders (FWO-Vlaanderen) (research project G021118 and research grant 1526418N). D.S. is a recipient of an FWO-SB PhD fellowship of the Research Foundation Flanders (FWO-Vlaanderen). Research in the laboratory of A.C.L. was supported by the Swedish Research Council, Grant 621-2013-4685.

## Author contributions

K.W. contributed by performing the ChIP-seq and RNA-seq experiments, protein purification, growth experiments and the data analysis; D.S. contributed by performing the qRT-PCR, EMSA, and footprinting experiments and data analysis; H.R.M. and C.L. contributed by performing the mutant constructions, protein purifications, EMSA experiments and the data analysis; L.L. contributed by preparing the *S. acidocaldarius* mutant strain, performing ChIP-qPCR and the data analysis; X.Z. contributed by performing the growth experiments, RNA extractions, and data analysis; F.S. contributed by performing protein purifications; C.B. and B.S. contributed in the study design and data analysis; K.V. contributed by performing protein crystallography and data analysis; A.C. L. and E.P. contributed by conceiving and designing the study, performing data analysis, and writing the paper. All authors approved of the paper.

## Additional information

**Competing interests:** The authors declare no competing interests.

