## [Peer Review File · Nature Communications]

Reviewers' comments:

Reviewer #1 (Remarks to the Author):

The manuscript, A bacterial-like FadR transcription factor regulates fatty acid metabolism in the archaeal model organism *Sulfolobus acidocaldarius*, reports the functional and structural properties of a bacterial-type TetR-family transcriptional regulator encoded by *Saci_1107*. The protein does not display outstanding primary structure similarity with their bacterial counterparts, but the protein structure reveals its similarity with previously characterized TetR-family FadR transcription regulators from *Bacillus* and *Thermus*. The authors proceed to show, by chromatin immunoprecipitation in combination with next-generation sequencing (ChIP-seq), the loci to which the protein binds (genes within the *Saci_1103*-*Sac1126* gene cluster). *Saci_1107* disruption, along with transcriptome analyses, confirms that these genes are controlled by the *Saci_1107* protein. Sequence analysis led to the prediction of the sequences recognized by the protein, and these were confirmed with EMSAs and, extensively with co-crystallization and footprint experiments. The authors proceed to further demonstrate that the *Saci_1107* protein dissociates from DNA when bound to medium- to long-chain acyl-CoA molecules. This is explained by elucidation of the lauroyl-CoA bound protein structure and its comparison with the unbound protein. The study is solid and complete, addressing the transcriptional regulation of genes involved in fatty acid metabolism from (i) structural, (ii) biochemical, and (iii) genetic approaches. The conclusions are in some cases justified by both in vitro and in vivo experiments. The results are of value to a broad readership, which should include those interested in microbial physiology, gene regulation, transcription, metabolism, protein structure, protein evolution and archaea. The manuscript is well written and was easy to follow despite the abundance of data. Studies on fatty acid metabolism in Archaea are just beginning, and this study should lay a foundation for progress on this topic. Specific comments are shown below for the authors to consider in order to further strengthen the manuscript.

Line 37: by a homolog of bacterial TetR-family FadR proteins: This might be an understatement, as the authors could not detect significant similarity at the primary structure level. Or is the reviewer misunderstanding the degree of similarity?

Line 72 and Fig. 1A: The authors seem to distinguish the 3-ketoacyl-CoA thiolases (acetyl-CoA acetyltransferases) with the beta-oxidation system enzymes. The concept may have recently changed, but thiolases are/were regarded as important members of the beta-oxidation system. There are also acetoacetyl-CoA thiolases that are important for chain elongation. Please consider how these enzymes should be described (It is true that the other three members are directly involved in oxidation).

Line 105: Co-crystallization of acyl-CoA with FadRSa suggests that it is a specific ligand of the protein. What should the reader refer to here? I suppose it is Figure 5, so perhaps the authors should insert a (see below) here.

Lines 114-116: High-enrichment and low-enrichment: Is it that the low-enrichment binding regions (peaks 3 and 4) in this cluster are still higher than other regions on the genome?

Lines 138-143: Can the authors illustrate this architecture somewhere in the main body of the manuscript? An illustration (at the end) or maybe a part of Supplementary Figure 5b would help the reader to understand what the authors mean here by dimer-of-dimer interaction mode.

Line 201: Have the authors searched for the TTGA..(X)8..TCAA sequence? In particular, is it found on any of the other intergenic regions in the gene cluster? Should we presume that the protein interacts with these regions? Can it be found on any of the promoters whose transcription levels increase with *Saci_1107* disruption?

Lines 215-217: Concerning the different conformation of the ligand, the results are clear and there does not seem to be any doubt about the different conformation. My question is, does the Saci_1107 protein dimer harbor the residues that would be expected to interact with the ligand in the dimer interface as in the bacterial proteins, or are these residues lost?

As the authors now demonstrate the function of the Saci_1107 protein, the distribution of the protein among archaea is interesting. Is the protein confined to Sulfolobus, or Crenarchaea or aerobic archaea? Does the distribution of the protein show any correlation with the distribution of the beta-oxidation proteins?

Supplementary Figure 5b: Is it possible to explain the colors?

Besides Supplementary Figure 5b, Supplementary Figure 12 might also be a candidate to move to the main body of the manuscript.

Typos

Journal of Molecular Biology, Molecular Microbiology, Nature Structural Biology should be abbreviated appropriately.

Reviewer #2 (Remarks to the Author):

The sole phenotype that links all archaea is the use of isoprenoid-based chains in their membranes to the exclusion of fatty acids. Despite not using fatty acids in their main metabolisms, many archaeal species are still exposed of fatty acids (as potential energy sources), or may generate fatty acids in small quantities in vivo. Much work remains to elucidate the biological importance of fatty acid metabolism in archaeal species.

Wang et al. structurally and functional dissect a TetR-family transcription regulator, FadR, from Sulfolobus acidocaldarius. The authors determine the crystal structure of FadRSa, both bound to DNA and substrate bound. Additionally, the authors determine the consensus sequence for FadRSa binding, and determine the binding sites across the genome with ChIP-seq. Transcriptomics, on WT and FadR deletion strains are also employed to determine the effect of FadRSa on genome-wide transcription regulation. A novel mechanism of acyl-CoA binding is identified in contrast to the substrate recognition mechanisms demonstrated for bacterial FadR homologues.

The description of a TetR-family transcription regulator in archaea is novel. The experiments presented in the manuscript are sound, technically correct, and there is little room to question the interpretations made by the authors. The work is solid, complete, and the manuscript is perhaps even overly descriptive of the results obtained. That said, the IMPACT of the current work on the field is unfortunately not large. Regulation of gene expression by FadR is clear, but the role of the gene cluster that may be involved (or may not be involved) in catabolic or anabolic fatty acid metabolism remains unclear. Thus, the biological importance of archaea fatty acid metabolism remains murky. The work presented is a great first step, but likely lacks the impact that is more typically associated with manuscripts published at Nature Communications. As written, the manuscript would be well received at J. Bact, Mol. Micro, or similar journals.

Major Points:

1. The authors repeatedly overstate how understanding FadRSa structure and mechanism of DNA binding will help to define the (putative) fatty acid metabolism of *S. acidocaldarius*. FadRSa regulation, dependent on acyl-CoA, of 23 genes predicted to be involved in fatty acid metabolism is described however the effects of deletion or modification FadRSa on fatty acid metabolism are not described. Do the authors have evidence of catabolic or anabolic fatty acid metabolism in *S. acidocaldarius*? The deletion of Sa_1107 demonstrates that this gene is not essential, but given the reported non-phenotypic result, what is the biological importance of archaeal fatty acid metabolism and its regulation? Does fatty acid content change within the cells deleted for FadR and how is this biologically important? These are essential questions to demonstrate the

importance of fatty acid function and regulation within archaea. Without them very little can be postulated regarding the importance of this regulator or the overall importance of this pathway.

2. The transcriptomics data suggests that 13 genes in the Saci_1103-Saci1126 gene cluster are differentially regulated in the absence of FadRSa, however within the gene cluster, there are four distinct binding sites. In the discussion, the authors suggest that long-range interactions and loop formation may be necessary for FadRSa repression. A demonstration of the overall mechanism of FadRSa repression of the gene cluster would greatly increase the impact of this manuscript.
3. Similar to the major point above, the suggestion that FadRSa is positioned to sterically inhibit RNAP recruitment to the promoter should be demonstrated.
4. Lines 271-272. Figure 5D seems to present exactly the OPPOSITE result, namely that longer chain acyl-CoA substrates alter DNA binding capacity of FadR.

Reviewer #3 (Remarks to the Author):

Archaea have isoprenoid-based chains in their membranes instead of fatty acid based chains, thus it has been unclear what role, if any, fatty acid metabolism plays in these organisms. The studies in this manuscript reveal a large cluster of genes that encode fatty acid enzymes regulated by a protein called FadR, suggesting at least a catabolic function for fatty acid metabolism in *S. acidocaldarius*. Specifically, this manuscript shows that the *S. acidocaldarius* FadR protein is a transcription regulator; ChIP analyses map the DNA binding sites of the protein and transcriptomic analyses reveal its regulon. Finally the authors determined crystal structures of FadR bound to DNA and lauroyl-CoA. The structures show FadR has a bacterial TetR fold. This is a really interesting and comprehensive study but there are several important issues that need to be addressed before consideration for publication.

1. An implied main finding from the study is that FadRsa binds target DNA sites as both dimers and dimer-of-dimers. The EMSA data seem to support this with stoichiometry estimates. However, the finding that the apo and DNA bound crystal structures revealed that the protein co-purifies with acyl-CoA molecules (which act as inducers) indicates that the protein they are using for EMSA is at least partially in the inactive, "induced" form. Therefore, DNA binding analyses with this protein would not be accurate. This is a problem because the issue of stoichiometry is critical for this paper and currently the EMSA studies are the only ones providing stoichiometry data. Because the DNA-bound structure also has some acyl-CoA bound subunits (see point 2 below) it does not unambiguously demonstrate dimer-of-dimer binding. Thus, the authors should use size exclusion chromatography (SEC) as a way to get at stoichiometry for the protein binding to DNA; they should use two DNA sites, one that the authors predict would bind FadR as a dimer and one as a dimer-of-dimer. This analyses could get around the non-homogeneous issue because they could use excess protein to ensure they saturate the DNA site with active (non-acyl-CoA bound) protein (which would isolate it away from the induced form) and use the resultant peak to get the MW and hence stoichiometry.

2. The authors also used EMSA to imply that FadR binding is cooperative. Again, this is a problem because their protein is apparently not fully in the DNA-binding active form. Also confusing is that they see two separate peaks in the EMSA (other TetR proteins that bind as cooperative dimers-of-dimers typically reveal a single shift upon binding). If the binding were fully cooperative they should only see one shift. They should tone down or not use the word cooperative to describe this binding.

3. The FadR-DNA structure has several caveats. First, the DNA site they used is apparently too small to bind a dimer-of-dimers and also one of the bound FadR dimers (the one in the center) is bound to acyl-CoA (inducer) and thus is not in a DNA binding active conformation, but an inducer bound conformation. Thus, this structure may not reflect the dimer-of-dimer bound structure in solution. Hence, the Asn37-Asn37' contact observed between dimers might also not be present in

the true dimer-of-dimer structure. They could test this by mutating Asn37 to see if that has an impact on dimer-of-dimer binding. But it looks like there is DNA bending between the DNA sites to permit binding of the inducer bound dimer. If the DNA is not bent, the dimer may thus not be positioned for such a contact. The author should really tone down this discussion and point out this caveat (they have some discussion on this in the supplementary data, but should move this discussion to the main text).

4. Also in regards to dimer-of-dimer binding, the authors say that Gly48 plays a key role in dimer-of-dimer binding via an electrostatic interaction between Gly48 and the base-specific N7 groups of guanines. First, I am unclear what base-specific N7 groups mean as adenine bases also has the N7 (donor) as guanines. Do they mean N6? They should show a figure of this key interaction to clarify their meaning.

5. The authors say that FadR binds the Saci_1123 DNA promoter site as a dimer and the only reason it does not bind this site as a dimer-of-dimer is that this DNA site is missing GC bps in the appropriate location (for Gly48 interaction) to allow that binding mode. Because they claim this is the only difference needed, the authors could demonstrate unequivocally that this is the case by generating the mutation and doing binding studies (or SEC) to show that the mutation indeed converts the site to one that binds FadR as a dimer-of-dimers rather than a dimer.

6. Along with the above points, Figure S5b may not be relevant as it assumes the dimer-of-dimer structure is the one observed when FadR is bound to a complete DNA site with both dimers in the DNA-bound form (and none in the inducer bound form).

7. The authors should show the electron density (omit map) for the bound lauroyl-CoA.

8. In Figure S6b, the Cu-OP footprint to the left seems to show a protected region not indicated in the figure above the one indicated. Can the authors explain why they did not include this as a protected region?

9. On page 8 they note that the DNA-bound and ligand bound structures superimpose with an rmsd of 1.01 Å. Is this the RMSD for overlaying just one subunit or both subunits of the dimer?

10. On page 4, can the authors provide the RMSDs for the superimpositions of the bacterial FadRs with FadRsa.

11. Figure 3g. Can the authors make the figure darker or bigger? The yellow letters in the figure are difficult to see.

Response letter NCOMMS-18-00633A “A TetR family transcription factor regulates fatty acid metabolism in the archaeal model organism *Sulfolobus acidocaldarius*” Wang *et al*

We gratefully acknowledge all reviewers for their valuable input, as it helped us greatly in improving the study and manuscript. Below you can find our point-to-point responses indicated in blue; line numbers refer to the revised version of the manuscript.

Reviewers' comments:

Reviewer #1 (Remarks to the Author):

The manuscript, A bacterial-like FadR transcription factor regulates fatty acid metabolism in the archaeal model organism *Sulfolobus acidocaldarius*, reports the functional and structural properties of a bacterial-type TetR-family transcriptional regulator encoded by *Saci_1107*. The protein does not display outstanding primary structure similarity with their bacterial counterparts, but the protein structure reveals its similarity with previously characterized TetR-family FadR transcription regulators from *Bacillus* and *Thermus*. The authors proceed to show, by chromatin immunoprecipitation in combination with next-generation sequencing (ChIP-seq), the loci to which the protein binds (genes within the *Saci_1103*-*Sac1126* gene cluster). *Saci_1107* disruption, along with transcriptome analyses, confirms that these genes are controlled by the *Saci_1107* protein. Sequence analysis led to the prediction of the sequences recognized by the protein, and these were confirmed with EMSAs and, extensively with co-crystallization and footprint experiments. The authors proceed to further demonstrate that the *Saci_1107* protein dissociates from DNA when bound to medium- to long-chain acyl-CoA molecules. This is explained by elucidation of the lauroyl-CoA bound protein structure and its comparison with the unbound protein.

The study is solid and complete, addressing the transcriptional regulation of genes involved in fatty acid metabolism from (i) structural, (ii) biochemical, and (iii) genetic approaches. The conclusions are in some cases justified by both *in vitro* and *in vivo* experiments. The results are of value to a broad readership, which should include those interested in microbial physiology, gene regulation, transcription, metabolism, protein structure, protein evolution and archaea. The manuscript is well written and was easy to follow despite the abundance of data. Studies on fatty acid metabolism in Archaea are just beginning, and this study should lay a foundation for progress on this topic. Specific comments are shown below for the authors to consider in order to further strengthen the manuscript.

1. Line 37: by a homolog of bacterial TetR-family FadR proteins: This might be an understatement, as the authors could not detect significant similarity at the primary structure level. Or is the reviewer misunderstanding the degree of similarity?

We thank the reviewer for pointing this out: indeed, the *S. acidocaldarius* FadR regulator should not be considered as homologous/orthologous to the bacterial TetR-like FadR regulators. Moreover, the two bacterial FadR regulators from *Thermus thermophilus* and *Bacillus* are shown not to be orthologous to

each other (*cfr.* Cuthbertson, L. & Nodwell, J.R. The TetR family of regulators. *Microbiol. Mol. Biol. Rev.* **77**, 440-475 (2013)). We have removed the statement from the abstract, upon rewriting this in response to comment 1 of reviewer 2. We have adapted the Discussion as follows: “*FadR_{Sac}*, of which we show that it displays structural similarities with bacterial TetR-like FadR regulators,... . Despite these similarities, there are pronounced differences between the archaeal FadR regulator and its bacterial counterparts.” (lines 360-363) and we have replaced the term “homologous” with “similar” and “homolog” with “counterpart” at several other locations throughout the text. To emphasize the lack of homology between the archaeal and bacterial TetR-like FadR regulators, we have also modified the title of the manuscript as follows: “A TetR family transcription factor regulates...” (lines 1-2) (in the previous version, it was mentioned: “A bacterial-like FadR transcription factor regulates...”).

2. Line 72 and Fig. 1A: The authors seem to distinguish the 3-ketoacyl-CoA thiolases (acetyl-CoA acetyltransferases) with the beta-oxidation system enzymes. The concept may have recently changed, but thiolases are/were regarded as important members of the beta-oxidation system. There are also acetoacetyl-CoA thiolases that are important for chain elongation. Please consider how these enzymes should be described (It is true that the other three members are directly involved in oxidation).

Indeed, there are two types of acetyl-CoA C-acetyltransferases: catabolic thiolases, functioning in the β -oxidation system, and anabolic thiolases, functioning in chain elongation in the mevalonate pathway in isoprenoid biosynthesis. According to reference 10 (Dibrova, D. V., Galperin, M. Y. & Mulkidjanian, A. Y. Phylogenomic reconstruction of archaeal fatty acid metabolism. *Environ. Microbiol.* **16**, 907–918 (2014)), many archaeal genomes encode multiple paralogs for the mevalonate pathway enzymes while several archaea also harbor multiple paralogs for bacterial-like β -oxidation thiolases. It is stated in this paper that the ketoacyl-CoA cleavage may be carried out by both types of enzymes: the “real” ketoacyl-CoA thiolase homologs, of which two are predicted to be encoded in the *S. acidocaldarius* genome, but also by mevalonate pathway acetyl-CoA C-acetyltransferases, of which eight paralogs are present in *S. acidocaldarius*. As such, the authors of reference 10 distinguished between acetyl-CoA C-acetyltransferases and β -oxidation enzymes and likewise, we have preserved that distinction in the Introduction, by naming acetyl-CoA C-acetyltransferases and β -oxidation enzymes separately. However, we have included a more elaborate explanation in the Introduction as follows: “*S. acidocaldarius* has an extensive gene cluster, comprising genes *Saci_1103* until *Saci_1126*, encoding homologs of the three β -oxidation enzymes acyl-CoA dehydrogenase, enoyl-CoA hydratase and hydroxyacyl-CoA dehydrogenase. Also, genes encoding members of the thiolase superfamily presumably catalyzing the last step of the β -oxidation cycle, i.e. ketoacyl-CoA thiolases as well as acetyl-CoA acetyltransferases were identified within the cluster¹⁰. In addition, genes encoding lipid degradation functions are present in this genomic region (Figure 1a).” (lines 73-79). The exact functional classification of acetyl-CoA C-acetyltransferases enzymes in *S. acidocaldarius*, including

those encoded in the *Saci_1103-Saci_1126* gene cluster (*Saci_1114* and *Saci_1121*), and whether or not they truly have a degradation function as suggested by the functional characterization of the FadR_{Sa} regulator, is unclear.

However, because it is beyond the scope of this paper to functionally classify the individual enzymes, we altered Figure 1a and 2a by classifying both β -oxidation enzymes and acetyl-CoA C-acetyltransferase enzymes more generally as “enzymes acting in fatty acid metabolism”. The same generalization was used in the newly prepared Figure 8b. Likewise, we have changed “ β -oxidation and acetyl-CoA acetyltransferase enzymes” to “ β -oxidation enzymes” in the abstract (lines 34-35) and we have removed the following sentence from the Discussion: “*Nevertheless, the presence of archaea-specific paralogs of acetyl-CoA C-acetyltransferase in combination with the β -oxidation enzyme-encoding genes, some even in the same operon, does not exclude the possibility that the pathway functions not only in a catabolic but also in an anabolic direction.*”.

3. Line 105: Co-crystallization of acyl-CoA with FadR_{Sa} suggests that it is a specific ligand of the protein. What should the reader refer to here? I suppose it is Figure 5, so perhaps the authors should insert a (see below) here.

Here, we refer to the “native” FadR_{Sa} structure that is explained in the same paragraph. To further clarify this to the reader, we have inserted a referral at the end of this sentence to Figure 1b, in which the ligand is shown and indicated (“heptanoyl-CoA”). Also, it is now emphasized that the co-crystallization occurred serendipitously: “*The unintended cocrystallization of acyl-CoA with FadR_{Sa} (Figure 1b) suggests that it is a specific ligand of the protein.*” (lines 111-112).

4. Lines 114-116: High-enrichment and low-enrichment: Is it that the low-enrichment binding regions (peaks 3 and 4) in this cluster are still higher than other regions on the genome?

This is not the case: as shown in Supplementary Table 2, the fold-enrichments for binding regions named peaks 3 and 4 are lower than those of two genomic ChIP regions outside the *Saci_1103-Saci_1126* gene cluster, although in the same range. We have adapted the text to make it clear to the reader that the observation of the highest enrichments taking place in the *Saci_1103-Saci_1126* gene cluster is solely based on the so-called high-enrichment binding regions (peaks 1 and 2): “*The **two** highest enrichments were observed within the *Saci_1103-Saci_1126* gene cluster. **More specifically, both** high-enrichment binding regions were located within the intergenic region of the divergently organized operon encoding the *fadR_{Sa}* gene itself and a putative esterase-encoding gene (peaks 1 and 2). **Within the *Saci_1103-Saci_1126* gene cluster**, two additional low-enrichment binding regions were observed within the coding sequence of gene *Saci_1115* and in the intergenic region separating a divergently encoded β -oxidation operon and a putative transcription factor gene, respectively (peaks 3 and 4).*” (lines 119-125).

5. Lines 138-143: Can the authors illustrate this architecture somewhere in the main body of the manuscript? An illustration (at the end) or maybe a part of Supplementary Figure 5b would help the reader to understand what the authors mean here by dimer-of-dimer interaction mode.

A more extensive analysis has been performed to further support the proposal of the dimer-of-dimer interaction mode (see response to comment 1 of reviewer 3). This is more elaborately described in a separate Results section in the revised manuscript (lines 162-222) and in Figure 4, including schematic illustrations of the architecture of the different stoichiometric binding modes in Figures 4a and 4f. As such, Supplementary Figure 4b (previously Supplementary Figure 5b) becomes less relevant for the illustration of the binding architecture and we do not consider it important enough to transfer it to the main body of the manuscript.

6. Line 201: Have the authors searched for the TTGA..(X)8..TCAA sequence? In particular, is it found on any of the other intergenic regions in the gene cluster? Should we presume that the protein interacts with these regions? Can it be found on any of the promoters whose transcription levels increase with Saci_1107 disruption?

We have performed the suggested binding motif searches with the FIMO tool of the MEME-suite for the entire *Saci_1103-Saci_1126* gene cluster (not only with the intergenic regions since one of the originally identified binding sites is an intragenic site), and found additional putative binding sites, which are however not confirmed for *in vivo* binding in the ChIP-seq analysis (see Supplementary Table 6, Figure 6a and lines 266-269): “*Besides the four experimentally identified binding sites, an in silico screening revealed three additional putative binding sites in the gene cluster, of which one is located within the ORF of Saci_1106 and is predicted to accommodate dimer-of-dimer binding*”. These additional binding motifs open the possibility that more genomic loci, which were not captured during the ChIP experiment, are involved in the formation of higher-order nucleoprotein complexes, as studied in the newly performed AFM experiments (see response to comment 2 of reviewer 2) (see Figure 6b-d and lines 278-288). We have also scanned the genomic regions encompassing the target genes identified to be downregulated in the RNA-seq analysis. Here, only few motifs were retrieved with minimal similarities, making it unlikely that they accommodate binding. This is explained on lines 239-242: “*The suggestion of this effect being indirect is corroborated by the prediction of only a limited number of putative FadR_{sa} binding sites in the genomic regions surrounding the downregulated genes, which are characterized by relative high P-values (> 1.00E⁻⁰⁵) (Supplementary Table 6) and which were not captured by ChIP-seq.*”

7. Lines 215-217: Concerning the different conformation of the ligand, the results are clear and there does not seem to be any doubt about the different conformation. My question is, does the Saci_1107

protein dimer harbor the residues that would be expected to interact with the ligand in the dimer interface as in the bacterial proteins, or are these residues lost?

Given the complete lack of homology in the C-terminal domain (see Figure 1d and Supplementary Table 1), we argue in our paper that there is no shared ancestry between FadR_{Sa} and bacterial FadR regulators: “*Despite these similarities, there are pronounced differences between the archaeal FadR regulator and the bacterial counterparts which point to a complete absence of shared ancestry.*” (lines 362-364). There is no conservation for any of the residues involved in ligand binding in the bacterial FadR_{Bh}, with the exception of an arginine in $\alpha 7$ (R118 in FadR_{Sa}), which is however not located close enough to the lauroyl-CoA ligand for it to establish a direct interaction. We have adapted Figure 1d to include annotation of the residues important for protein function in bacterial FadR_{Bh} and included the following information in the text: “*Although the nature of these ligand-interaction residues (polar residues for CoA interactions, hydrophobic residues for side chain interactions) is similar to those in bacterial FadR regulators, they are not homologous as shown on a structure-based sequence alignment (Figure 1d).*” (lines 312-315).

8. As the authors now demonstrate the function of the Saci_1107 protein, the distribution of the protein among archaea is interesting. Is the protein confined to Sulfolobus, or Crenarchaea or aerobic archaea? Does the distribution of the protein show any correlation with the distribution of the beta-oxidation proteins?

This is indeed a very interesting question that relates the observations in our work towards other archaeal species or lineages. We have performed BLAST analyses with the FadR_{Sa} protein sequence and retrieved several putative homologs with high sequence identities and similarities. These were found in all other *Sulfolobus* species, several other Crenarchaeota and Euryarchaeota and in the newly discovered Marsarchaeota phylum. Interestingly, also in these cases the FadR regulator is encoded in genomic regions that harbor fatty acid and lipid degradation genes including those encoding β -oxidation enzymes, suggesting the existence of similar FadR-mediated long-range repression mechanisms. These new findings are presented in Figure 8 and described in the text in the Results section “*Occurrence of FadR in archaea*” (lines 343-355).

9. Supplementary Figure 5b: Is it possible to explain the colors?

This information has been added to the figure legend (now Supplementary Figure 4b).

10. Besides Supplementary Figure 5b, Supplementary Figure 12 might also be a candidate to move to the main body of the manuscript.

Supplementary Figure 5b (now Supplementary figure 4b) has not been moved to the main body of the manuscript for reasons explained in our answer to comment 5. Supplementary Figure 12b has been

transferred to Figure 7 in the manuscript by switching it with a figure panel showing Met displacement (Figure 7c and Supplementary Figure 15a).

Typos

Journal of Molecular Biology, Molecular Microbiology, Nature Structural Biology should be abbreviated appropriately.

This has been corrected.

Reviewer #2 (Remarks to the Author):

The sole phenotype that links all archaea is the use of isoprenoid-based chains in their membranes to the exclusion of fatty acids. Despite not using fatty acids in their main metabolisms, many archaeal species are still exposed of fatty acids (as potential energy sources), or may generate fatty acids in small quantities in vivo. Much work remains to elucidate the biological importance of fatty acid metabolism in archaeal species.

Wang et al. structurally and functional dissect a TetR-family transcription regulator, FadR, from *Sulfolobus acidocaldarius*. The authors determine the crystal structure of FadRSa, both bound to DNA and substrate bound. Additionally, the authors determine the consensus sequence for FadRSa binding, and determine the binding sites across the genome with ChIP-seq. Transcriptomics, on WT and FadR deletion strains are also employed to determine the effect of FadRSa on genome-wide transcription regulation. A novel mechanism of acyl-CoA binding is identified in contrast to the substrate recognition mechanisms demonstrated for bacterial FadR homologues.

The description of a TetR-family transcription regulator in archaea is novel. The experiments presented in the manuscript are sound, technically correct, and there is little room to question the interpretations made by the authors. The work is solid, complete, and the manuscript is perhaps even overly descriptive of the results obtained. That said, the IMPACT of the current work on the field is unfortunately not large. Regulation of gene expression by FadR is clear, but the role of the gene cluster that may be involved (or may not be involved) in catabolic or anabolic fatty acid metabolism remains unclear. Thus, the biological importance of archaea fatty acid metabolism remains murky. The work presented is a great first step, but likely lacks the impact that is more typically associated with manuscripts published at Nature Communications. As written, the manuscript would be well received at J. Bact, Mol. Micro, or similar journals.

Major Points:

1. The authors repeatedly overstate how understanding FadRSa structure and mechanism of DNA binding will help to define the (putative) fatty acid metabolism of *S. acidocaldarius*. FadRSa regulation, dependent on acyl-CoA, of 23 genes predicted to be involved in fatty acid metabolism is described however the effects of deletion or modification FadRSa on fatty acid metabolism are not described. Do

the authors have evidence of catabolic or anabolic fatty acid metabolism in *S. acidocaldarius*? The deletion of Sa_1107 demonstrates that this gene is not essential, but given the reported non-phenotypic result, what is the biological importance of archaeal fatty acid metabolism and its regulation? Does fatty acid content change within the cells deleted for FadR and how is this biologically important? These are essential questions to demonstrate the importance of fatty acid function and regulation within archaea. Without them very little can be postulated regarding the importance of this regulator or the overall importance of this pathway.

In line with this remark we have now performed cultivation in the presence of fatty acids (hexanoate and butyrate) and demonstrated for the first time the growth of *S. acidocaldarius* cells with short-chain fatty acids as sole carbon and energy source. Furthermore, a phenotypic effect of deleting *fadr_{Sa}* during growth on hexanoate, but not butyrate, is demonstrating thereby revealing a link between the *Saci_1103-Saci_1126* gene cluster and its FadR_{Sa}-mediated regulation on one hand and fatty acid catabolism on the other hand. These new results are presented in Figure 5c and Supplementary Figure 11 and in the revised text as follows: “*The observed transcriptional regulation of the Saci_1103-Saci_1126 gene cluster strongly suggests that FadR_{Sa} is implicated in the regulation of fatty acid and lipid metabolism. Since it was previously observed that simultaneous deletion of both esterase-encoding genes in the gene cluster (Saci_1105 and Saci_1116) led to a phenotype lacking the ability to perform tributyrin hydrolysis²⁹, we performed a similar phenotypic assay with the fadr_{Sa} deletion mutant (Supplementary Figure 11a). Despite the higher expression levels of both esterase genes in the fadr_{Sa} deletion mutant (Figure 5a-b), we did not observe a difference in time-dependent halo formation upon growth on tributyrin (Supplementary Figure 11a). In contrast, upon growing S. acidocaldarius in a liquid medium containing hexanoate as a sole carbon and energy source, the fadr_{Sa} deletion mutant displayed a significantly higher growth rate in exponential growth phase with respect to the isogenic WT strain (doubling times T_ds of 20.5 and 26.3 hours, respectively; Figure 5c). As this effect was not observed during growth on the shorter-chain butyrate (Supplementary Figure 11b), it correlates to fatty acid chain length. These experiments demonstrate that S. acidocaldarius is capable of degrading fatty acids to sustain growth and that this catabolic metabolism is at least partly catalyzed by enzymes encoded in the Saci_1103-Saci_1126 gene cluster. Furthermore, the FadR_{Sa} regulator represses this catabolic fatty acid metabolism as its deletion, thereby causing a derepression of the gene cluster, results in a faster growth rate (Figure 5c).*” (lines 246-262). We have adapted the Discussion accordingly: “*The finding that FadR_{Sa} represses the Saci_1103-Saci_1126 gene cluster and that it is responsive to acyl-CoA molecules acting as inducers in vitro strongly suggests that intracellularly **present** acyl-CoA **molecules** cause a derepression and thus higher transcriptional expression of the gene cluster in vivo. **The observation that deletion of the regulator causes cells to display a faster growth on hexanoate as sole energy and carbon source demonstrates that the β -oxidation enzymes encoded in this gene cluster minimally have a degradation function; this is in line with the logic behind the regulatory strategy. A catabolic function of the β -oxidation enzymes is also in agreement with the function of the co-regulated esterase***

*enzymes encoded by Saci_1105 and Saci_1116, which enable cells to grow on lipids*²⁹. ***Fatty acid oxidation adds to the chemoorganotrophic capabilities of Sulfolobus spp. that appear more important than the originally described chemolithotrophic sulfur-oxidizing metabolism***³⁹. *A full picture of the functioning of fatty acid metabolism in Sulfolobus, and whether the enzymes encoded in the Saci_1103-1126 function only the catabolic or also anabolic direction, awaits the biochemical and genetic characterization of the enzymes.*" (lines 396-408).

Unfortunately, despite many attempts to cultivate *S. acidocaldarius* in the presence of long-chain fatty acids this appeared to be impossible because of the toxicity of detergents required to solubilize the fatty acids. Furthermore, the reviewer's suggestion of quantitatively analyzing fatty acid content in the wildtype and FadR_{Sa} knockout strains appears technically challenging and not promising to yield an informative result: a previous study demonstrated that fatty acids can be detected in cell extracts of the related *Sulfolobus solfataricus* grown in similar conditions as in our study with GC-MS and NMR, but that amounts are too low for quantification (Hamerly, T., Tripet, B., Wurch, L., Hettich, R.L., Podar, M., Bothner, B. & Copié, V. Characterization of fatty acids in Crenarchaeota by GC-MS and NMR. *Archaea* **2015** 472726 (2015)). Given the relative low repression folds observed in RNA-seq and qRT-PCR, differences in intracellular fatty acid content in wildtype and FadR_{Sa} knockout cells will likely be limited and difficult to quantify.

We agree with the reviewer that we should not claim that our study enables to clearly define the role of fatty acids in archaeal metabolism or physiology. Rather, as mentioned by reviewer 1, this study lays a foundation for progress on this topic and will aid in the design of future studies, for example with the aim of characterizing the individual enzymes encoded by the *Saci_1103-Saci_1126* gene cluster, which is beyond the scope of the current study. The potential impact of our paper should not only be situated in helping to unravel the role of fatty acid metabolism in archaea, but even more so in the paradigmatic shift in understanding how transcriptional repressors function in archaea and how a prototypical prokaryotic transcription factor family has evolved differently in bacteria and archaea: i) this is the first archaeal repressor shown to cause gene silencing through long-range interactions and DNA loop formation instead of through direct interactions with the basal transcription machinery (see also response to comment 2) and ii) besides similarities in the global structure the archaeal TetR-like FadR regulator employs distinct mechanisms from the bacterial counterparts (acyl-CoA interaction mode, DNA-binding mode, transcription regulatory mechanism). In line with these arguments, we have altered the text at several positions to emphasize the novelty of the mechanisms of the regulator and to tone down hypotheses with respect to fatty acid metabolism. We have significantly re-written the abstract limiting attention on fatty acid metabolism and emphasizing the novelty of the regulatory mechanisms. We have added additional experimental sections on the characterization of the regulator, including a section on structural effects on DNA (see our response to comment 2), which further support acyl-CoA-dependent FadR_{Sa}-mediated regulation to be a regulatory mechanism that has not yet been described in archaea or in bacteria. We have rewritten the Discussion, thereby limiting interpretations and postulations with

respect to fatty acid metabolism to a single paragraph (lines 396-408) and ending the Discussion with a paragraph regarding the putative regulatory mechanism (lines 417-428). Finally, we have enlarged the potential impact of our work by adding a Results section in which we relate these findings to other archaea (see response to comment 8 of reviewer 1).

2. The transcriptomics data suggests that 13 genes in the Saci_1103-Saci1126 gene cluster are differentially regulated in the absence of FadRSa, however within the gene cluster, there are four distinct binding sites. In the discussion, the authors suggest that long-range interactions and loop formation may be necessary for FadRSa repression. A demonstration of the overall mechanism of FadRSa repression of the gene cluster would greatly increase the impact of this manuscript.

We are grateful for this consideration, as it inspired us to perform qualitative atomic force microscopy analysis of FadR_{Sa} binding *in vitro* to a synthetic DNA fragment that mimics the gene cluster and harbors the four binding sites. FadR_{Sa} caused clear DNA condensation and looping, an observation that supports our initial hypothesis. This is an important observation as it goes beyond the paradigm of archaeal repressors and it adds to the mechanistic differences between archaeal and bacterial TetR-like FadR regulators. We have added these new results to the manuscript in Figure 6b-d and in lines 278-288. Furthermore, we have adapted the abstract: “*Atomic force microscopy imaging of the architecture of FadR_{Sa}-DNA complexes suggest that indirect gene silencing is achieved by condensing the entire genomic region through long-range protein-protein interactions, which represents a non-conventional regulatory mechanism for TetR-like regulators and for archaeal repressors.*” (lines 35-38) and we have discussed this new finding in the Discussion (lines 384-395 and lines 417-425).

3. Similar to the major point above, the suggestion that FadRSa is positioned to sterically inhibit RNAP recruitment to the promoter should be demonstrated.

We have removed this hypothesis from the Discussion. In addition, we have performed EMSAs of FadR_{Sa} binding to the *fadR_{Sa}* promoter region in combination with the basal transcription factors TBP and TFB that demonstrate that FadR_{Sa} does not inhibit binding of these factors, but rather stimulates their binding by forming a stable supershifted complex (Supplementary Figure 12). This is discussed in the Results section: “*As an exception, the fadR_{Sa} control region harbors a binding site just downstream of the transcription start site (TSS). For this target, it is shown that FadR_{Sa} binding stimulates the interaction with basal transcription factors TATA binding protein (TBP) and transcription factor B (TFB) (Supplementary Figure 11), pointing to a direct repression mechanism occurring at later stages of transcription initiation than during TBP and TFB recruitment.*” (lines 273-277).

4. Lines 271-272. Figure 5D seems to present exactly the OPPOSITE result, namely that longer chain acyl-CoA substrates alter DNA binding capacity of FadR.

Here, we describe ligand-binding specificity in comparison with bacterial FadR regulators. To clarify this and avoid misunderstanding, we have rewritten the concerned sentence: “As a consequence, while the extent of the ligand-response effect still correlates with the acyl chain length, *FadR_{Sa}* appears more sensitive to shorter chain-length acyl-CoAs **in comparison to bacterial FadR.**” (lines 370-372).

Reviewer #3 (Remarks to the Author):

Archaea have isoprenoid-based chains in their membranes instead of fatty acid based chains, thus it has been unclear what role, if any, fatty acid metabolism plays in these organisms. The studies in this manuscript reveal a large cluster of genes that encode fatty acid enzymes regulated by a protein called FadR, suggesting at least a catabolic function for fatty acid metabolism in *S. acidocaldarius*. Specifically, this manuscript shows that the *S. acidocaldarius* FadR protein is a transcription regulator; ChIP analyses map the DNA binding sites of the protein and transcriptomic analyses reveal its regulon. Finally the authors determined crystal structures of FadR bound to DNA and lauroyl-CoA. The structures show FadR has a bacterial TetR fold. This is a really interesting and comprehensive study but there are several important issues that need to be addressed before consideration for publication.

1. An implied main finding from the study is that FadR_{Sa} binds target DNA sites as both dimers and dimer-of-dimers. The EMSA data seem to support this with stoichiometry estimates. However, the finding that the apo and DNA bound crystal structures revealed that the protein co-purifies with acyl-CoA molecules (which act as inducers) indicates that the protein they are using for EMSA is at least partially in the inactive, “induced” form. Therefore, DNA binding analyses with this protein would not be accurate. This is a problem because the issue of stoichiometry is critical for this paper and currently the EMSA studies are the only ones providing stoichiometry data. Because the DNA-bound structure also has some acyl-CoA bound subunits (see point 2 below) it does not unambiguously demonstrate dimer-of-dimer binding. Thus, the authors should use size exclusion chromatography (SEC) as a way to get at stoichiometry for the protein binding to DNA; they should use two DNA sites, one that the authors predict would bind FadR as a dimer and one as a dimer-of-dimer. This analyses could get around the non-homogeneous issue because they could use excess protein to ensure they saturate the DNA site with active (non-acyl-CoA bound) protein (which would isolate it away from the induced form) and use the resultant peak to get the MW and hence stoichiometry.

To extend binding stoichiometry analysis, we have performed the suggested SEC experiments with FadR_{Sa} and 45-bp DNA probes harboring either the *fadR_{Sa}* or *Saci_1106* operator sequence. These new results are presented in Figure 4a and lines 163-181. They provide additional proof that FadR_{Sa} forms a different type of complex with each of the operators. Estimates of apparent molecular weights suggest that the complex with the *Saci_1106* operator has a single dimer bound while the *fadR_{Sa}* operator has two dimers bound. The calculated MWs of the complexes do not accurately equal the sum of the MWs of the individual components but this can be linked to the conformation of the DNA molecule that is

significantly altered upon binding (underscored by the hyperreactivity effects observed in footprinting experiments (Figure 4c, Supplementary Figures 7 and 8) and by the architecture of the complexed regions in AFM images (Figure 6c)), which is expected to affect SEC migration velocity differently in a bound *versus* unbound state of the DNA. In contrast, FadR_{Sa} protein remains in a globular shape when bound. This reasoning is explained in detail in the text. SEC experiments were performed in an excess of FadR_{Sa} protein (40 nmol protein *versus* 1 nmol DNA), as suggested by the reviewer to circumvent possible effects of a partial population of the protein being in a ligand-bound state lacking DNA-binding activity. However, following reasons allow us to exclude the possibility that a significant part of the protein is inactive in solution: i) we have performed additional SEC analyses with lower protein:DNA molar ratios and observed that almost all free protein is titrated away in the favor of forming the protein-DNA complex. These new results are presented in Supplementary Figure 6 and in the text (lines 176-181): “SEC experiments with lower protein:DNA molar ratios indicate that the entire amount of FadR_{Sa} in the preparation is capable of binding DNA (Supplementary Figure 6). This excludes the possibility that a subpopulation of the protein is in a ligand-induced state lacking DNA-binding activity as suggested by the observation of acyl-CoA cocrystallizing with the protein in the apo crystal structure (Figure 1b), assuming that acyl-CoA binding causes DNA dissociation like in bacterial FadR regulators.”; ii) in the apo crystal structure, only one of the two subunits harbored acyl-CoA in its ligand-binding pocket and these were short-chain acyl-CoAs, which have been shown to minimally induce dissociation of protein-DNA complexes (Figure 7d). It can thus be concluded that all *in vitro* binding experiments were performed with a homogenous population of active FadR_{Sa} protein molecules, and that our interpretations on binding characteristics throughout the study can be considered valid.

2. The authors also used EMSA to imply that FadR binding is cooperative. Again, this is a problem because their protein is apparently not fully in the DNA-binding active form. Also confusing is that they see two separate peaks in the EMSA (other TetR proteins that bind as cooperative dimers-of-dimers typically reveal a single shift upon binding). If the binding were fully cooperative they should only see one shift. They should tone down or not use the word cooperative to describe this binding.

We beg to disagree with the reviewer on this point. First, as explained in our response to comment 1, we have now provided additional proof that the protein is fully in a DNA-binding active form. Second, protein-DNA binding curves obtained by EMSA are routinely used for the analysis of binding cooperativity using Hill curve fitting, with the Hill coefficient being a quantitative measure of cooperativity (n=1: no cooperativity, n>1: positive cooperativity). The observation that the Hill coefficient is 1.4 indicating a lack of cooperativity in case of FadR_{Sa} binding to the *Saci_1123* operator, while it is 5.0 for binding to the *fadR_{Sa}* operator indicating positive cooperativity (Supplementary Figure 3 and Figure 2d) is an important element in the distinction of the two binding modes and we therefore disagree to remove this from the manuscript. Third, it is not true that in EMSA analyses intermediate complexes are never observed in case of binding cooperativity as this depends on the thermodynamic

characteristics of the interaction, as well as the used EMSA conditions and protein concentrations (for example, in the EMSAs shown in Figure 4b and in Supplementary Figure 3b, the intermediate FadR_{Sa}-DNA complex B1 is hardly detectable because of the transient nature of the complex). In this context, we would like to refer the reviewer to a previous paper in which we dissect binding cooperativity by analyzing the formation of intermediate complexes in EMSAs (Peeters, E., van Oeffelen, L., Nadal, M., Forterre, P. & Charlier D. A thermodynamic model of the cooperative interaction between the archaeal transcription factor Ss-LrpB and its tripartite operator DNA. *Gene* **524**, 330-340 (2013)). Finally, it is a righteous remark that intermediate complexes were never observed in previous studies with bacterial dimer-of-dimer binding TetR proteins: this is yet another mechanistic difference between bacterial and archaeal TetR regulators.

3. The FadR-DNA structure has several caveats. First, the DNA site they used is apparently too small to bind a dimer-of-dimers and also one of the bound FadR dimers (the one in the center) is bound to acyl-CoA (inducer) and thus is not in a DNA binding active conformation, but an inducer bound conformation. Thus, this structure may not reflect the dimer-of-dimer bound structure in solution. Hence, the Asn37-Asn37' contact observed between dimers might also not be present in the true dimer-of-dimer structure. They could test this by mutating Asn37 to see if that has an impact on dimer-of-dimer binding. But it looks like there is DNA bending between the DNA sites to permit binding of the inducer bound dimer. If the DNA is not bent, the dimer may thus not be positioned for such a contact. The author should really tone down this discussion and point out this caveat (they have some discussion on this in the supplementary data, but should move this discussion to the main text).

First, we would like to mention that we have attempted FadR_{Sa}-DNA cocrystallization with a variety of DNA probes differing in length, namely 15-, 17-, 19-, 21-, 23-, 25- and 42-mer oligonucleotide duplexes harboring different sections of the *fadR_{Sa}* operator. Besides the crystal structure that was obtained with a 21-mer DNA molecule and that is presented in the manuscript, we have also obtained structures with shorter probes (see Figure shown below). However, these only harbored a single dimer or two dimers, each of which only had a single subunit bound. As we consider these structures to contain less information with respect to the biologically relevant complex than the 21-mer complex structure, we decided not to present them in the manuscript. The use of DNA probes longer than 21-mer did not yield crystals, possibly due to flexibility of the DNA ends in the complex. Second, we notice that the reviewer misinterpreted the composition of the cocrystal structure presented in Figure 3a. This structure harbors two individual DNA molecules with the center of the structure not representing protein-induced DNA bending of a single DNA molecule, but representing two DNA molecules being close to each other but entirely disconnected. As such, the relative positioning of the two major groove segments that interact with the central ligand-bound FadR_{Sa} dimer is only possible because they belong to distinct DNA molecules and is probably not expected to occur by bending a single DNA molecule. This underlies the mechanism of ligand-induced DNA dissociation. Third, we argue that the structure of a single DNA

molecule (*e.g.* XY) interacting with an entire dimer (*e.g.* AB) and one of the subunits of the central ligand-bound dimer (*e.g.* E) does reflect the dimer-of-dimer type complex formed in solution for the following reasons: i) the nature of the protein-DNA interactions of subunit E on one hand and subunits A and B on the other hand are very similar to each other with the crucial Gly48-G interaction being established by both subunits E and B. The binding of a second dimer, represented by subunit E in the structure, to the complex is solely determined by the presence of a G-C on the position with which Gly48 of subunit E interacts. This is now proven by the new experiment presented in Figure 4f, in which the mutation of the corresponding position in the *Saci_1123* binding site yields dimer-of-dimer instead of dimer complex formation (see also our response to comment 5). As such, we can assume that the overlapping inverted repeats that are contacted in the cocrystal structure are identical to those contacted in the EMSA and footprinting binding experiments and that the position of subunit E reflects that of one of the interacting subunits of the second dimer that binds in a dimer-of-dimer binding mode (interacting with the “left” operator defined in Figure 4d); ii) as suggested by the reviewer, we have performed site-directed mutagenesis of the Asn37 residue and subjected a FadR_{Sa}^{N37A} protein preparation to a binding analysis. Using a probe that forms dimer-of-dimer complexes with the wildtype protein, it is demonstrated that a larger fraction of the dimer complex B1 is formed at the expense of the dimer-of-dimer complex B2 (Figure 4f). This indicates a diminished cooperativity and demonstrates that the Asn37-mediated protein-protein interaction is not only established in the cocrystal structure, but also in the dimer-of-dimer complex observed in EMSA; iii) the observation that the central FadR_{Sa} dimer in the structure has an acyl-CoA-bound induced conformation does not interfere with the translation of the observed contacts and molecular conformations in the crystal structure towards the biologically relevant dimer-of-dimer structure as we only consider a single subunit (*e.g.* subunit E). Indeed, ligand binding induces conformational changes mainly involving the relative positioning of two monomers within a dimer (as explained in the manuscript in Results section “Molecular mechanism of ligand response”). We have adapted the text at several locations to include the results with the Asn37 mutant, to avoid misinterpretation of the presented cocrystal structure and to explain our reasoning of why this cocrystal structure reflects the biologically relevant dimer-of-dimer complex more clearly to the reader (lines 198-222).

Figure. FadR_{Sa}-DNA cocrystal structures with 15-mer DNA molecules harboring the *fadR_{Sa}* operator binding site (not presented in the manuscript).

4. Also in regards to dimer-of-dimer binding, the authors say that Gly48 plays a key role in dimer-of-dimer binding via an electrostatic interaction between Gly48 and the base-specific N7 groups of guanines. First, I am unclear what base-specific N7 groups mean as adenine bases also has the N7 (donor) as guanines. Do they mean N6? They should show a figure of this key interaction to clarify their meaning.

As requested we have added a figure of the zoom of the interaction between Gly48 and G14 (Figure 4e). A hydrogen bond is established with the N7 group of the guanine base and there are no other interactions taking place between the concerned amino acid and base. We thank the reviewer for this comment, as it points out that the Gly48-G interaction that we showed to be crucial for dimer-of-dimer complex formation is indeed not specific for guanines, but could theoretically also be established with adenine bases. Inspection of all (putative) dimer-of-dimer binding sites shows that none of them harbors an adenine on the corresponding position, but we have altered our conclusion as dimer-of-dimer binding being determined by a Gly48-purine interaction instead of a Gly48-G interaction at several locations throughout the text.

5. The authors say that FadR binds the Saci_1123 DNA promoter site as a dimer and the only reason it does not bind this site as a dimer-of-dimer is that this DNA site is missing GC bps in the appropriate location (for Gly48 interaction) to allow that binding mode. Because they claim this is the only difference needed, the authors could demonstrate unequivocally that this is the case by generating the mutation and doing binding studies (or SEC) to show that the mutation indeed converts the site to one that binds FadR as a dimer-of-dimers rather than a dimer.

We have performed this experiment and the results corroborate the hypothesis of the G-C/C-G bps being the sole sequence determinants of dimer-of-dimer binding. This new result is shown in Figure 4f and explained in the text (lines 209-212): *“This reasoning is underscored by the observation that the introduction of a G-C and C-G bp at the indicated positions of the Saci_1123 operator causes the formation of two instead of one nucleoprotein complex (Figure 4f).”*

6. Along with the above points, Figure S5b may not be relevant as it assumes the dimer-of-dimer structure is the one observed when FadR is bound to a complete DNA site with both dimers in the DNA-bound form (and none in the inducer bound form).

We would like to refer to our response to comment 3 to support the assumption that the conformation of the portion of the cocrystal structure with the AB dimer and E subunit of the EF dimer reflects that of the biologically relevant dimer-of-dimer complex. Supplementary Figure 4b (previously Supplementary Figure 5b) depicts the relative orientation of these two dimers by showing only this part of the structure. This is now also explained in the figure legend. As this figure is only used to depict the relative orientation of the two dimers with respect to each other, and not of individual monomers within

a dimer, the ligand-bound state of the EF dimer should not influence this depiction. Indeed, ligand binding induces conformational changes mainly involving the relative positioning of two monomers within a dimer (as explained in the manuscript in Results section “Molecular mechanism of ligand response”).

7. The authors should show the electron density (omit map) for the bound lauroyl-CoA.

This is now shown as Supplementary Figure 13a.

8. In Figure S6b, the Cu-OP footprint to the left seems to show a protected region not indicated in the figure above the one indicated. Can the authors explain why they did not include this as a protected region?

We do not consider this region as protected by protein for two reasons: i) this protection is not observed in the corresponding region when performing the experiment with the bottom strand labelled (see right-hand panel of the figure), while all other footprinting protection zones are confirmed by protection observed for both DNA strands; ii) a nonspecific decrease in band intensity for the larger fragments can be explained by the chemical footprinting reaction conditions leading to excessive DNA cleavage in this specific experiment. This is supported by the complete absence of unreacted DNA molecules on top of the autoradiograph image, while these are generally observed in the other footprinting experiments. We have provided an explanation about this in the legend of this figure (now Supplementary Figure 7b).

9. On page 8 they note that the DNA-bound and ligand bound structures superimpose with an rmsd of 1.01 Å. Is this the RMSD for overlaying just one subunit or both subunits of the dimer?

This RMSD value was obtained for a superposition with both subunits; we have added this information to the legend of Supplementary Figure 15b.

10. On page 4, can the authors provide the RMSDs for the superimpositions of the bacterial FadRs with FadRsa.

This is now provided in the text (lines 98-102): “*Although BLAST analyses initially did not reveal which bacterial regulators could be considered as potential functional homologs for the protein encoded by Saci_1107, a superposition revealed high structural similarity with the previously characterized TetR-family FadR transcription regulators in Bacillus sp., FadR_{Bs} (RMSD = 4.23 Å) and FadR_{Bh}^{21,26} (RMSD=5.88 Å), and Thermus thermophilus, FadR_{Tt}²⁴ (RMSD = 11.85 Å) (Figure 1c).*”

11. Figure 3g. Can the authors make the figure darker or bigger? The yellow letters in the figure are difficult to see.

Unfortunately, it was difficult to enlarge this figure panel as the entire figure contains many panels (now Figure 4). However, we have darkened the yellow letters aiming to improve readability (now Figure

4d).

Reviewers' comments:

Reviewer #2 (Remarks to the Author):

The revised manuscript is an improvement on the original and addresses a broad subject area of general interest to the archaeal and bacterial communities. The writing is less tedious, and the manuscript employs a host of techniques that convincingly demonstrate the main thrust of the text; e.g. FadR almost certainly acts as a repressor of genes involved in fatty acid biosynthesis in *S. acidocaldarius*. The supplemental materials are exhaustive and most experimentation is carried out with precision. Unfortunately, the devil is in the details, some aspects of the experimentation are not fully-supported by the data presented. The manuscript is currently so over interpreted that it is not suitable for publication. Areas of concern are addressed below:

1 - Does fatty acid catabolism occur in *S. solfataricus* and can such catabolism support growth? The revised manuscript shows growth is possible on hexanoate (but not butyrate) and this is a critical piece of information in support of publication. Although the Saci1103-Saci1126 gene cluster was not targeted for deletion (which presumably could and should be done), the interpretations for this cluster supporting fatty acid catabolism are sufficient. The faster growth rate is not an obvious phenotype of the deletion strain however, as introduction of hexanoate should, in theory, completely disrupt repressive activities of FadR, making the strain behave effectively as a FadR deletion strain. Comment should be made to correct this interpretation or to explain how only partial derepression is achieved in media supplemented with hexanoate. The penultimate paragraph of the discussion clarifies this situation more appropriately and this level of interpretation should be placed directly in the results section.

2 - Are the structures of FadR, app and acyl-CoA bound, and DNA-bound biologically relevant? The crystallographically resolved structures of FadR (without DNA) are not in question. However, the co-structures with DNA may or may not represent biologically relevant structures. To many reviewers this may be a major concern, as the details of binding and repression might be illuminated by more biologically relevant structures, but the extensive mutagenesis that correlates with FadR function is sufficient in my opinion to detail the bulk of FadR-FadR interactions and FadR-DNA interactions. FadR clearly binds operator sequences, this binding is influenced by acyl-CoA binding, and the mechanism of acyl-CoA binding is seemingly 1) well-resolved, and 2) unique to the archaeal protein. However, the depth of comments, and the unnecessary stress of importance of demonstrating that the archaeal and bacterial proteins function differently is likely only critical to a small community of researchers involved in this regulator family, and thus discussion of such could be significantly trimmed throughout. There does not appear to be sufficient evidence at this time to truly describe the binding as dimers vs. dimers of dimers, or some mix of each. The manuscript would benefit from a less combative tone in describing the likely DNA-bound structures. The biology of regulation is likely much more important than the differences in binding modes between domains.

3 - How does FadR binding regulate gene expression within a large gene cluster with many transcription units, only some of which have defined operator sequences? How do operator sequences that are not adjacent/overlapping with core promoter elements influence transcription??

It is important to mention that the authors acknowledge that the regulation of this gene cluster is likely more complex than the regulation provided by FadR alone (they do such in the discussion), but the results section is over-interpreted and thus misleading regarding what can be interpreted from the available experimentation. The results must be completely rewritten with regards to the biological interpretations of potential FadR looping.

Regardless of the mechanism of binding, it is clear that FadR can bind to multiple locations in the genome, and given that mutagenesis supports FadR-FadR contacts, it is plausible that distant

operator sequences are brought together in three-dimensional space to assist in regulating the entire Saci1103-1126 gene cluster. The authors add two important experiments. On the positive, Sup Figure 12 demonstrates basal transcription factors not only can bind, but whose binding is enhanced/stabilized by FadR binding. The results of this experiment are clear and suggest that an alternative mechanism of repression is responsible at fadR control region in Fragment 2.

The authors now include AFM analyses of FadR interactions with long fragments of DNA, and here the conclusions that can be convincingly recovered drop precipitously. No controls nor statistics are provided. Before drawing any conclusions regarding the possibility of repression via looping, the AFM experimentation should be repeated with long DNA fragments wherein the operator sequences were scrambled to disrupt FadR binding. This control is critical to any interpretations possible from such an AFM analysis. Controls with FadR variants that fail to bind DNA or that interfere with dimer-dimer interactions should also be included. The introduction of acyl-CoAs should likewise be added as controls. Without the controls, the current explanations cannot support more than the most tepid suggestion of DNA looping control gene expression. The discussion and results from the AFM experimentation should be removed entirely until the proper controls are available.

If we assume that the AFM supports looping, then:

- 1 - does genome mining of other species that encode putative catabolic operons and FadR homologues/orthologues also support binding of FadR at distant locations with the operons? This assumes that operator sequences can be identified.
- 2 - What percentage of complexes that fall into class I, II, III, or IV? This information would be critical in supporting looping, as if class I dominates, limits are placed on interpretations of rare events.
- 3 - Defined transcriptional studies wherein DNA looping hinders transcription typically result from very small loops that either block recruitment of RNAP, block open complex formation via introduction of positive supercoiling, or inhibit promoter escape by destabilizing the initial transcription elongation complex. The putative looping that may be possible via FadR would result in loops that are kb long, and there is no obvious reason why such constraints would hinder transcription from multiple promoters with the gene cluster. The available data certainly does not support the strong language in the discussion of a novel and new mechanism of regulating archaeal transcription.
- 4 - Given the genetic accessibility of the model species, individual operator sequences could and should be modified to clarify their potential role in mediated three-dimensional structure formation that somehow limits transcription of the gene cluster.
- 5 - Given the ploidy of *S. acidocaldarius*, the authors have not mentioned how multiple genomes would be controlled and how looping between genomes would be prevented. The AFM results (as they are currently interpreted) permit binding of many DNA molecules, and this would certainly impact in vivo regulation.

In conclusion, the manuscript is improved, but too over interpreted and too poorly supported for publication at this time. The biological story of fatty acid catabolism in archaeal species will have FadR as a central component, and the structures of the protein (apo and bound) are valuable, but too many unknowns regarding DNA-binding, putative looping, additional regulatory factors, etc. remain for publication in a high-profile journal. Finally, Multiple mentions to the "first, to our knowledge" should be removed. These statements are presumably added for impact, but they read as pompous and add little value.

Reviewer #3 (Remarks to the Author):

Wang et al. have performed a number of new experiments in this revision to address reviewer's comments and concerns. I am generally in favor of publication but have a few remaining issues that could be addressed.

1. I had no confusion about the protein-DNA structure from the standpoint that what the authors captured was a complex that is formed between DNA substrates (ie dimers are not bound to the same DNA substrate). That was one issue that raised the question of whether the complex was physiologically relevant. That and the finding that one of the dimers had substrate/inducer bound. The authors point out that the dimer with CoA bound in this complex makes the same or similar contacts to the DNA as the non-CoA bound dimer, which is good evidence towards it being relevant despite being bound to a possible inducer. But I could not find anywhere (unless I missed it) where they discussed the distance between HTHs of the DNA bound and CoA bound forms. This measurement typically denotes the difference between DNA binding active forms of TetR proteins and their induced forms (the latter typically having increased distances between HTH of subunits of the dimer compared to dimers that are bound to DNA). They also don't mention distances between HTH of the lauroyl-CoA bound form and the DNA bound dimer – they just indicate lauroyl-CoA binding "opens up" the dimer. Providing distances of the DNA bound form (without CoA bound), the CoA bound and DNA bound and the lauroyl-CoA bound forms would be informative. That being said, the authors do perform DNA binding studies of requested mutant proteins that support their model. However, to TetR aficionados it would be informative to know these distances.

2. The finding that Gly48 contacts N7 of guanine is somewhat confusing in their argument that this interaction is highly base specific (as an adenine also has N7). However, it could be due to indirect readout. It is not clear to me in just looking at the figure but if the TpG (where the G is contacted by Gly48) is somewhat unstacked to allow this interaction this may be a form of indirect readout. This is because TpG base pairs (and in general pyrimidine-G base steps are more prone to unstacking. Indeed, there is a well known protein-DNA interaction called a YpG interaction made by arginine side chains to the guanine that is possible precisely because of this tendency of this base pair to unstack. They may be seeing a similar thing but where the gly is able to contact the guanine due to this unstacking (see Lamoureux et al J Mol Biol 2004 for details). If this is not the case, then it remains unclear why there is this preference.

3. I do not feel that the authors needed to add the AFM data to enhance the impact. The issue I have with the AFM data is it is very difficult to see where the proteins are in these images. Is there a way they can point them out in each image?

Response letter 2nd revision NCOMMS-18-00633A “A TetR family transcription factor regulates fatty acid metabolism in the archaeal model organism *Sulfolobus acidocaldarius*” Wang *et al*

Below you can find our point-to-point responses indicated in blue; line numbers refer to the revised version of the manuscript.

Editorial comments:

Both referees feel that the AFM data do not seem to add much to the article, and we agree with this view; therefore, ***please remove the AFM data from the paper***. In addition, please ensure that any overinterpretations (as highlighted by referee #2) are removed or suitably rephrased throughout the manuscript. Furthermore, we ask that you provide additional details to the structural analyses as requested by referee #3 (e.g. distance between HTHs of the DNA bound and CoA bound forms, etc). This is not to say that we consider any other points raised by our reviewers less important.

We have removed the AFM data and their discussion from the revised version of the manuscript. Two paragraphs that were presented in the same section as the AFM data have been moved to other appropriate locations in the manuscript. The paragraph on the *in silico* identification of additional binding sites (previously requested by reviewer 1) has now been moved to the section on “Genome-wide DNA interaction map of FadR_{Sa}” and the paragraph on the binding experiment with basal transcription factors (previously requested by reviewer 2) is now presented in the section “Determination of the FadR_{Sa} regulon”. For the other editorial comments, we would like to refer to our responses below.

Reviewers' comments:

Reviewer #2 (Remarks to the Author):

The revised manuscript is an improvement on the original and addresses a broad subject area of general interest to the archaeal and bacterial communities. The writing is less tedious, and the manuscript employs a host of techniques that convincingly demonstrate the main thrust of the text; e.g. FadR almost certainly acts as a repressor of genes involved in fatty acid biosynthesis in *S. acidocaldarius*. The supplemental materials are exhaustive and most experimentation is carried out with precision. Unfortunately, the devil is in the details, some aspects of the experimentation are not fully-supported by the data presented. The manuscript is currently so over interpreted that it is not suitable for publication. Areas of concern are addressed below:

1 - Does fatty acid catabolism occur in *S. solfataricus* and can such catabolism support growth? The revised manuscript shows growth is possible on hexanoate (but not butyrate) and this is a critical piece of information in support of publication. Although the Saci1103-Saci1126 gene cluster was not targeted for deletion (which presumably could and should be done), the interpretations for this cluster supporting fatty acid catabolism are sufficient. The faster growth rate is not an obvious phenotype of the deletion strain however, as introduction of hexanoate should, in theory, completely disrupt repressive activities

of FadR, making the strain behave effectively as a FadR deletion strain. Comment should be made to correct this interpretation or to explain how only partial derepression is achieved in media supplemented with hexanoate. The penultimate paragraph of the discussion clarifies this situation more appropriately and this level of interpretation should be placed directly in the results section.

The observation of a slight difference in growth rate between the FadR_{Sa} deletion strain and the wildtype strain in a medium with hexanoate as a sole carbon and energy source can be explained by short-chain acyl-CoA molecules having relative smaller effects on the inhibition of DNA binding by FadR_{Sa} than medium- and longer-chain acyl-CoA molecules (see Figure 6d). Even at a hexanoyl-CoA concentration of 100 μM only a small fraction of the protein-DNA complexes are dissociated *in vitro*, indicating that an extracellularly applied hexanoate concentration of 2 mM presumably does not lead to a full dissociation of FadR_{Sa} from the DNA *in vivo*. We have added a sentence in the text to further clarify this: “*The observation of this difference can be explained by hexanoate only causing a partial FadR_{Sa}-mediated derepression given the relative short chain length of these acyl-CoA molecules (see below, “FadR_{Sa}-ligand interactions”).*” (lines 271-274). In our opinion, it is less appropriate to include a complete interpretation at the level that is presented in the Discussion (currently the last paragraph, lines 381-403), as suggested by the reviewer, at this position in the manuscript because experimental results describing the effects of ligand binding on functioning of FadR_{Sa} are presented only in a later section in the text, making it confusing for the reader to follow such an interpretation at this earlier stage.

2 - Are the structures of FadR, app and acyl-CoA bound, and DNA-bound biologically relevant? The crystallographically resolved structures of FadR (without DNA) are not in question. However, the co-structures with DNA may or may not represent biologically relevant structures. To many reviewers this may be a major concern, as the details of binding and repression might be illuminated by more biologically relevant structures, but the extensive mutagenesis that correlates with FadR function is sufficient in my opinion to detail the bulk of FadR-FadR interactions and FadR-DNA interactions. FadR clearly binds operator sequences, this binding is influenced by acyl-CoA binding, and the mechanism of acyl-CoA binding is seemingly 1) well-resolved, and 2) unique to the archaeal protein. However, the depth of comments, and the unnecessary stress of importance of demonstrating that the archaeal and bacterial proteins function differently is likely only critical to a small community of researchers involved in this regulator family, and thus discussion of such could be significantly trimmed throughout. There does not appear to be sufficient evidence at this time to truly describe the binding as dimers vs. dimers of dimers, or some mix of each. The manuscript would benefit from a less combative tone in describing the likely DNA-bound structures. The biology of regulation is likely much more important than the differences in binding modes between domains.

To de-emphasize the importance of the observation of functional differences between the bacterial and archaeal regulators, we have removed the terms “Strikingly,” and “Curiously” (lines 37 and 280) from

the text. Furthermore, we have removed any comparisons between the bacterial and archaeal regulators in terms of regulatory mechanism (“*Not only do the ligand-interaction and DNA-binding modes distinguish FadR_{Sa} from bacterial TetR-like FadR regulators, the achievement of gene silencing of a large genomic region through long-range interactions affecting the structure of genomic DNA is unique*”) (see also our response to comment 3). In contrast, the comparative discussion of the functioning of bacterial and archaeal FadR with regards to ligand interactions (presented on lines 353-365) and DNA-binding mechanisms (presented on lines 191-195 and 220-224) is to our opinion a critical element of the manuscript for following reasons: i) functional aspects of elaborately studied bacterial TetR-like FadR regulators provide a “framework” for the correct interpretation of experimental observations for this novel member of the TetR family; ii) the hypothesis that archaeal FadR_{Sa} has originated from convergent evolution rather than that it was acquired through horizontal gene transfer from bacteria, which is based on the elaborately described functional and structural differences between archaeal and bacterial regulators as presented in the manuscript, is an outcome of our work that has implications for the further study of the biological role of these regulators and of fatty acid metabolism in archaea. Therefore, we find it important to leave the conclusion of convergent evolution in the last sentence of the Abstract (lines 37-40) and the first section of the Discussion (lines 348-365) in the manuscript. However, to comply with the reviewer’s request of minimizing this discussion, we have shortened the description of the comparison of bacterial and archaeal FadR in the remainder of the Discussion as follows: “*As a consequence, ~~while the extent of the ligand-response effect still correlates with the acyl chain length, FadR_{Sa} appears more sensitive to shorter chain length acyl CoAs in comparison to bacterial FadR. A has a different ligand specificity in comparison to bacterial FadR, which is expected to have consequences for the biological function and to~~ Vice versa, it can be postulated that the differently evolved ligand binding specificities reflect different biological roles of fatty acids for cellular physiology, as they are crucial for membrane lipid homeostasis in bacteria but have different as yet unknown functions in bacteria and archaea.*” (lines 360-362). The sentence “*Bacterial TetR-family regulators are subdivided in one of two subclasses depending on their DNA-binding mode.*” has been removed from the Discussion.

Regarding the reviewer’s statements “the unnecessary stress of importance of demonstrating that the archaeal and bacterial proteins function differently is likely only critical to a small community of researchers involved in this regulator family” and “The biology of regulation is likely much more important than the differences in binding modes between domains.” we would like to respond by mentioning that it is difficult to objectively assign potential impacts to individual aspects of the protein’s functioning as this is likely to be biased by the background and research interests of individual readers. Furthermore, the potential impact of the study of regulators belonging to the TetR family is demonstrated by the high number of citations for the latest review paper published on this topic, Cuthbertson, L. & Nodwell, J.R. The TetR family of regulators. *Microbiol. Mol. Biol. Rev.* **2013**; 77: 440-475 (109 citations, source: Web of Science), which seemingly contradicts that only a small

community of researchers is involved in the study of this regulator family. In this context, we would also like to cite Reviewer 1 in the first peer review report of our manuscript: “The results are of value to a broad readership, which should include those interested in microbial physiology, gene regulation, transcription, metabolism, protein structure, protein evolution and archaea.”

3 - How does FadR binding regulate gene expression within a large gene cluster with many transcription units, only some of which have defined operator sequences? How do operator sequences that are not adjacent/overlapping with core promoter elements influence transcription??

It is important to mention that the authors acknowledge that the regulation of this gene cluster is likely more complex than the regulation provided by FadR alone (they do such in the discussion), but the results section is over-interpreted and thus misleading regarding what can be interpreted from the available experimentation. The results must be completely rewritten with regards to the biological interpretations of potential FadR looping.

Regardless of the mechanism of binding, it is clear that FadR can bind to multiple locations in the genome, and given that mutagenesis supports FadR-FadR contacts, it is plausible that distant operator sequences are brought together in three-dimensional space to assist in regulating the entire Saci1103-1126 gene cluster. The authors add two important experiments. On the positive, Sup Figure 12 demonstrates basal transcription factors not only can bind, but whose binding is enhanced/stabilized by FadR binding. The results of this experiment are clear and suggest that an alternative mechanism of repression is responsible at fadR control region in Fragment 2.

The authors now include AFM analyses of FadR interactions with long fragments of DNA, and here the conclusions that can be convincingly recovered drop precipitously. No controls nor statistics are provided. Before drawing any conclusions regarding the possibility of repression via looping, the AFM experimentation should be repeated with long DNA fragments wherein the operator sequences were scrambled to disrupt FadR binding. This control is critical to any interpretations possible from such an AFM analysis. Controls with FadR variants that fail to bind DNA or that interfere with dimer-dimer interactions should also be included. The introduction of acyl-CoAs should likewise be added as controls. Without the controls, the current explanations cannot support more than the most tepid suggestion of DNA looping control gene expression. The discussion and results from the AFM experimentation should be removed entirely until the proper controls are available.

On request of both reviewers (see also comment 3 of reviewer 3) and the editor, we have removed the AFM data and their discussion from the revised version of the manuscript. Therefore, the main focus of the manuscript is placed on unravelling structure and function of archaeal FadR with respect to ligand binding, DNA binding, regulon identification and physiological role and not on the characterization of the regulation mechanism. A more detailed AFM analysis, including the suggested controls and a quantitative analysis, thus falls beyond the scope of the current manuscript, and could be the subject of a future stand-alone study of long-range DNA interactions by FadR_{Sa}.

If we assume that the AFM supports looping, then:

- 1 - does genome mining of other species that encode putative catabolic operons and FadR homologues/orthologues also support binding of FadR at distant locations with the operons? This assumes that operator sequences can be identified.
- 2 - What percentage of complexes that fall into class I, II, III, or IV? This information would be critical in supporting looping, as if class I dominates, limits are placed on interpretations of rare events.
- 3 - Defined transcriptional studies wherein DNA looping hinders transcription typically result from very small loops that either block recruitment of RNAP, block open complex formation via introduction of positive supercoiling, or inhibit promoter escape by destabilizing the initial transcription elongation complex. The putative looping that may be possible via FadR would result in loops that are kb long, and there is no obvious reason why such constraints would hinder transcription from multiple promoters with the gene cluster. The available data certainly does not support the strong language in the discussion of a novel and new mechanism of regulating archaeal transcription.
- 4 - Given the genetic accessibility of the model species, individual operator sequences could and should be modified to clarify their potential role in mediated three-dimensional structure formation that somehow limits transcription of the gene cluster.
- 5 - Given the ploidy of *S. acidocaldarius*, the authors have not mentioned how multiple genomes would be controlled and how looping between genomes would be prevented. The AFM results (as they are currently interpreted) permit binding of many DNA molecules, and this would certainly impact *in vivo* regulation.

By removing the AFM data from the manuscript, most of the above suggestions, although very valuable, have become irrelevant in the context of the current version of the manuscript:

1. It is an interesting suggestion to predict FadR binding sites in other archaeal species that harbor a FadR homolog: we have performed an *in silico* screening of the gene clusters of closely related species also belonging to the *Sulfolobus* genus for sequences that are similar to a consensus motif based on the identified FadR_{sa} binding site sequences. This search resulted in the prediction of autoregulatory sites and additional sites at distant locations, either in intergenic regions or in open reading frames, revealing the possibility that FadR homologs function similarly in these related species. This new result is now presented in the Results section (lines 342-345) and is presented in Figure 7b.
2. By removing the AFM analysis, this question is not relevant anymore.
3. Statements of DNA looping occurring between the distantly bound FadR_{sa} molecules and the concerned paragraph regarding “a novel mechanism of regulating archaeal transcription” have been removed from the manuscript.
4. Although it is an interesting suggestion to delete operator sequences *in vivo*, it will be difficult to assign transcriptional changes to the deletion of these binding site sequences and the absence

of FadR_{Sa} binding, since some of these are located within open reading frames or close to basal transcriptional and translational elements (the autoregulatory site itself). Furthermore, the study of the *in vivo* three-dimensional structure of the gene cluster appears technically challenging. Above all, by removing the AFM data and the discussion regarding long-range interactions and the putative regulatory mechanism of FadR_{Sa} from the manuscript, the suggested experiment is not relevant anymore for the current version.

5. Crenarchaeota, including *Sulfolobus acidocaldarius*, are monoploid (Soppa, J. Ploidy and gene conversion in archaea. *Biochem. Soc. Trans.* **39**, 150-154), harboring a single copy of the genome except during the cell cycle phase following DNA replication. Whether or not FadR_{Sa}-mediated intermolecular interactions take place between the two daughter chromosomes just after DNA replication is a technically highly challenging question and falls out of scope for this manuscript given that we have removed results and discussion regarding long-range interactions and the putative regulatory mechanism of FadR_{Sa}.

In conclusion, the manuscript is improved, but too over interpreted and too poorly supported for publication at this time. The biological story of fatty acid catabolism in archaeal species will have FadR as a central component, and the structures of the protein (apo and bound) are valuable, but too many unknowns regarding DNA-binding, putative looping, additional regulatory factors, etc. remain for publication in a high-profile journal. Finally, Multiple mentions to the "first, to our knowledge" should be removed. These statements are presumably added for impact, but they read as pompous and add little value. We have removed the mentions of "first" and "to our knowledge": The previous sentences in the Introduction "*We aimed at unravelling the function and mode of action of this regulator, named FadR_{Sa}, by performing structural, biochemical, genetic and genomic analyses. To our knowledge, this is the first report of an acyl-CoA-responsive transcriptional regulator in an archaeal microorganism of which functional insights enable us to formulate hypotheses on the role of fatty acid metabolism in these organisms.*" have been replaced by "***We aimed at performing structural, biochemical, genetic and genomic analyses of this regulator, named FadR_{Sa}, thereby unveiling the function and mode of action of an acyl-CoA responsive transcriptional regulator in an archaeal microorganism.***" (lines 79-82). The following sentence in the Discussion has been adapted as well: "*FadR_{Sa} of which we show that it displays structural similarities with bacterial TetR-like FadR regulators, is ~~to our knowledge the first characterized~~ an archaeal member of the widespread prokaryotic TetR family.*" (lines 350-351).

Reviewer #3 (Remarks to the Author):

Wang et al. have performed a number of new experiments in this revision to address reviewer's comments and concerns. I am generally in favor of publication but have a few remaining issues that could be addressed.

1. I had no confusion about the protein-DNA structure from the standpoint that what the authors captured was a complex that is formed between DNA substrates (ie dimers are not bound to the same DNA substrate). That was one issue that raised the question of whether the complex was physiologically relevant. That and the finding that one of the dimers had substrate/inducer bound. The authors point out that the dimer with CoA bound in this complex makes the same or similar contacts to the DNA as the non-CoA bound dimer, which is good evidence towards it being relevant despite being bound to a possible inducer. But I could not find anywhere (unless I missed it) where they discussed the distance between HTHs of the DNA bound and CoA bound forms. This measurement typically denotes the difference between DNA binding active forms of TetR proteins and their induced forms (the latter typically having increased distances between HTH of subunits of the dimer compared to dimers that are bound to DNA). They also don't mention distances between HTH of the lauroyl-CoA bound form and the DNA bound dimer – they just indicate lauroyl-CoA binding “opens up” the dimer. Providing distances of the DNA bound form (without CoA bound), the CoA bound and DNA bound and the lauroyl-CoA bounds forms would be informative. That being said, the authors do perform DNA binding studies of requested mutant proteins that support their model. However, to TetR aficionados it would be informative to know these distances.

While the distances between HTH motifs of two monomeric subunits are mentioned for lauroyl-CoA-bound and DNA-bound FadR_{Sa} in the section “Molecular mechanism of ligand response” (lines 318-321), we have also included this distance for the central ligand-bound dimer in the protein-DNA cocrystal structure in the main text of the manuscript (previously, this was only mentioned in Supplementary Text 3): “*this is reflected by the distance between the two $\alpha 3$ helices being 45.3 Å for this central dimer EF versus an average of 37.0 Å for the flanking dimers (Figure 3a; Supplementary Text 3).*” (lines 326-327).

2. The finding that Gly48 contacts N7 of guanine is somewhat confusing in their argument that this interaction is highly base specific (as an adenine also has N7). However, it could be due to indirect readout. It is not clear to me in just looking at the figure but if the TpG (where the G is contacted by Gly48) is somewhat unstacked to allow this interaction this may be a form of indirect readout. This is because TpG base pairs (and in general pyrimidine-G base steps are more prone to unstacking. Indeed, there is a well known protein-DNA interaction called a YpG interaction made by arginine side chains to the guanine that is possible precisely because of this tendency of this base pair to unstack. They may be seeing a similar thing but where the gly is able to contact the guanine due to this unstacking (see

Lamoureux et al J Mol Biol 2004 for details). If this is not the case, then it remains unclear why there is this preference.

This is indeed a very plausible explanation for the sole observation of guanines at this binding site interaction position; furthermore, this view is enforced by all Gly48-contacted guanines being preceded by either a thymidine or cytosine (the latter being the case for the upstream half of the central binding motif in the *fadR_{5a}* operator (Figure 4d)). In the new version of the manuscript, we have added this explanation and the mentioned reference as follows: “*Given that adenines also have an N7 group, the base specificity of the Gly48-guanine interaction is possibly explained by an indirect readout of the preceding thymidine or cytosine residue in the light of YpG base pair steps being more prone to unstacking and commonly involved in sequence-specific protein-DNA interactions in a combined direct and indirect readout*³².” (lines 210-213). In agreement to this new element in the interpretation, we have changed the expression “Gly-purine interaction” to “Gly-guanine interaction” throughout the text (lines 222 and 366).

3 .I do not feel that the authors needed to add the AFM data to enhance the impact. The issue I have with the AFM data is it is very difficult to see where the proteins are in these images. is there a way they can point them out in each image?

We have removed the AFM data and their discussion from the revised version of the manuscript, also in response to the editorial comments and to comment 3 of reviewer 2.